# Co-aggregation with Apolipoprotein E modulates the function of Amyloid-β in Alzheimer's disease

Zengjie Xia [1,2,12], Emily E. Prescott[3,12], Agnieszka Urbanek[3,12], Hollie E. Wareing[3], Marianne C. King[3], Anna Olerinyova[3], Helen Dakin [1,2,4], Tom Leah[3], Katy A. Barnes[3], Martyna M. Matuszyk[3], Eleni Dimou [1,2], Eric Hidari [1,2], Yu P. Zhang[1,2], Jeff Y. L. Lam [1,2], John S. H. Danial[1,2,5], Michael R. Strickland [6,7], Hong Jiang[6], Peter Thornton[8], Damian C. Crowther [8], Sohvi Ohtonen [9], Mireia Gómez-Budia [9], Simon M. Bell[3,10], Laura Ferraiuolo [3], Heather Mortiboys [3,10,11], Adrian Higginbottom[3,10], Stephen B. Wharton[3,10], David M. Holtzman [6], Tarja Malm [9], Rohan T. Ranasinghe[1,2] ✉, David Klenerman [1,2] ✉ & Suman De [3,10,11] ✉

Which isoforms of apolipoprotein E (apoE) we inherit determine our risk of developing late-onset Alzheimer's Disease (AD), but the mechanism underlying this link is poorly understood. In particular, the relevance of direct interactions between apoE and amyloid-β (Aβ) remains controversial. Here, single-molecule imaging shows that all isoforms of apoE associate with Aβ in the early stages of aggregation and then fall away as fibrillation happens. ApoE-Aβ co-aggregates account for ~50% of the mass of diffusible Aβ aggregates detected in the frontal cortices of homozygotes with the higher-risk *APOE4* gene. We show how dynamic interactions between apoE and Aβ tune disease-related functions of Aβ aggregates throughout the course of aggregation. Our results connect inherited *APOE* genotype with the risk of developing AD by demonstrating how, in an isoform- and lipidation-specific way, apoE modulates the aggregation, clearance and toxicity of Aβ. Selectively removing non-lipidated apoE4-Aβ co-aggregates enhances clearance of toxic Aβ by glial cells, and reduces secretion of inflammatory markers and membrane damage, demonstrating a clear path to AD therapeutics.

Inherited variation in the sequence of apolipoprotein E (apoE) is the greatest genetic risk factor for late-onset Alzheimer's Disease (AD), which accounts for ~95% of all AD cases[1]. The *APOE* gene has three alleles: *APOE2*, *APOE3*, and *APOE4*. Compared to the most common *APOE3* form, *APOE2* is neuroprotective[2], while *APOE4* increases AD risk; *APOE4* homozygotes are 15-fold more susceptible to AD than *APOE3* homozygotes[3–5], and disease begins several years earlier in *APOE4* carriers than those with *APOE3* or *APOE2*[3].

Attempts to establish how apoE influences AD risk have focused on its effect on the pathological amyloid-beta (Aβ) peptide[1,6]. Deposits of Aβ aggregates in the central nervous system (CNS) are a hallmark of AD[7], and disturbed Aβ homeostasis is one of the earliest events in AD pathogenesis[8,9]. This dysfunction induces synaptic and axonal damage as well as tau seeding and spreading, leading to neurodegeneration in AD[10–12]. There is compelling evidence that apoE4 enhances Aβ pathology[6]: carriers of the *APOE4* gene have more Aβ deposits in their

CNS than non-carriers[1,13], exhibit amyloid positivity earlier in life[14], and experience a faster-growing Aβ burden[15]. The isoforms of the apoE protein seem to differentially affect the levels of Aβ in the CNS, which may explain their differential AD risk profiles. Lipidation of apoE may also play a role in AD. Although apoE is mostly lipidated in the human CNS[6,16], poorly- or non-lipidated apoE increases Aβ pathology[17], and the risk of developing AD in both mouse[18] and human[19]. Observational studies show that apoE influences the relationship between Aβ and cognitive decline in AD[20–22]. However, it is unclear how the different isoforms and lipidation states of apoE affect Aβ aggregation, clearance, and aggregate-induced neurotoxicity at the molecular level.

The faster Aβ plaque deposition in carriers of *APOE4*[15,23] has led to several hypotheses for apoE4's molecular role in AD. One possibility is that apoE4 promotes more aggregation of Aβ than the other isoforms by interacting directly with Aβ when the two meet in the extracellular space. The discovery of co-deposited apoE in AD amyloid plaques provided early circumstantial evidence for this idea[16,24,25], but biophysical studies are equivocal: apoE can either speed up or slow down Aβ aggregation in vitro, depending on the conditions[26]. Targeting non-lipidated apoE in amyloid plaques with an antibody can significantly reduce Aβ pathology in transgenic mouse models[16,24] as well as reduce Aβ-induced tau seeding and spreading[27]. However, studies disagree on whether apoE and Aβ associate significantly in brain tissue[19,28]. The role of apoE-Aβ interactions in Aβ clearance is also unclear: while apoE might traffic Aβ out of the interstitial brain fluid[23,29], other work suggests that apoE and Aβ instead compete for clearance-mediating receptors[28].

In this study, we used single-molecule imaging to monitor how apoE affects the form, function, and clearance of Aβ aggregates at different stages of maturity. We show that apoE influences the structure and composition of Aβ aggregates during oligomerization, and that apoE-Aβ interactions fine-tune disease-relevant functions of Aβ in an isoform- and lipidation-specific manner. Our work identifies a role for transient, non-lipidated apoE-Aβ co-aggregates in modulating Aβ deposition and neurotoxicity, and thereby connects *APOE* genotype with the risk of developing sporadic AD.

## Results

### Early-stage apoE-Aβ Co-aggregates Form Isoform-independently, but Accumulate Isoform-dependently in AD Brains

We began our study by investigating how apoE interacts with Aβ along its aggregation pathway to fibrils. Aggregating Aβ comprises a dynamic, heterogeneous mixture of species with different sizes, shapes, and properties, and small sub-populations may disproportionately contribute to AD[30,31]. We reasoned that apoE could influence AD risk by interacting with Aβ aggregates that are transient and/or rare, which might explain why previous attempts to study association by taking snapshots of the bulk mixture have painted an inconsistent picture. We therefore aggregated Aβ42 in vitro at a concentration (4 μM) that would give rise to fibrils, in the presence and absence of near-physiological concentrations of each non-lipidated and lipidated apoE isoform (~ 80 nM)[32,33].

Assaying fibril formation using ThT fluorescence confirmed that all reactions produced fibrils and that the presence of non-lipidated apoE slowed aggregation, while the presence of lipidated apoE sped the reaction up (Fig. 1A, Figure S1, S2); lipid particles are known to accelerate the aggregation of amyloidogenic proteins by providing a surface for primary nucleation[34–36]. In order to sample the heterogeneity within each aggregation pathway, we characterized individual aggregates at different stages of the reaction. We imaged each aggregation reaction at the end of the lag phase ($t_1$), the middle of the growth phase ($t_2$), and the plateau phase ($t_3$), using single-molecule pull-down (SiMPull)[37] (Fig. 1B). In this assay (Fig. 1C), Aβ42 is captured using a surface-tethered 6E10 antibody and imaged using two-color total internal reflection fluorescence (TIRF) after adding primary

detector antibodies for Aβ (Alexa-Fluor-647-labeled 6E10) and apoE (Alexa-Fluor-488-labeled EPR19392) (Figure S4). Using the same monoclonal antibody to sandwich Aβ aggregates renders unreacted monomers undetectable because they only contain one epitope.

Characterizing individual aggregates rather than their ensemble average allowed us to extract properties of the heterogeneous population including size, shape, and composition; aggregates containing apoE and Aβ42 should be colocalized in both detection channels. These images (Fig. 1D–F) revealed that all types of apoE coaggregate with Aβ42 in the early stages ($t_1$ and $t_2$) of aggregation, but that co-aggregates disappear as the reaction reaches completion. Co-immunoprecipation of $t_1$ and $t_3$ aggregates with Aβ-antibody-coated beads followed by western blotting for both Aβ and apoE confirmed that apoE is absent from $t_3$ co-aggregates rather than merely inaccessible to antibodies (Fig. 2). There was no significant isoform dependence in the extent of colocalization, whether based on the aggregate number (Fig. 1G), or intensity (Fig. 1H). These findings are independent of the antibody used to detect apoE (Figure S5), and there is no colocalization when isotype-control detection antibodies are used (Figures S6 and S7), suggesting minimal contributions from non-specific binding. Using the intensity as a proxy for the amount of protein present in each aggregate suggests that ~75-100% of aggregate mass at the end of the lag phase is in co-aggregates, which falls to ~30%-60% in the growth phase, and ~0% by the plateau phase (Fig. 1H). Although non-lipidated apoE colocalizes with Aβ42 at slightly higher levels, the main effect of lipidation was on the apparent size of co-aggregates (Fig. 1I): those containing non-lipidated apoE (mean diameter ~500–900 nm) were much larger than those containing lipidated apoE (mean diameter ~200-250 nm or less; in diffraction-limited imaging, any aggregates smaller than the diffraction limit will appear in this range) or no apoE.

Our finding that non-lipidated apoE forms large early-stage co-aggregates with Aβ supports the idea that apoE in this form can stabilize intermediate species formed early on in aggregation and thus inhibit Aβ fibrillation[38]. Because we used apoE in sub-stoichiometric amounts (Aβ:apoE 50:1), it is unlikely to slow down aggregation by sequestering Aβ monomers. ApoE more likely interacts with Aβ42 in its earliest stages of aggregation; apoE did not associate with pre-formed Aβ42 aggregates (Figure S8). The high fluorescence intensities of these early-stage co-aggregates indicate high effective protein concentrations, which may seem incompatible with decreased fibrillization rates. However, it has recently been shown that locally concentrating Aβ42 in condensates can significantly slow its aggregation[39]. The fact that fibrils have shed all associated apoE might indicate that elongating heteronuclei is less energetically favorable than elongating homonuclei. Importantly, the lack of any isoform dependence means that co-aggregation alone cannot explain the *APOE* dependence of AD.

To determine whether these apoE-Aβ co-aggregates are disease-relevant, we isolated Aβ aggregates from the postmortem frontal cortices of six AD patients (Method section - Table 1). We chose a method that gently extracts diffusible aggregates by soaking[40], in preference to tissue homogenization, which extracts insoluble, predominantly fibrillar aggregates that are mostly inert[41]. We imaged the extracts from three homozygous *APOE4* and three homozygous *APOE3* AD patients using SiMPull (Fig. 3A). All samples yielded Aβ aggregates both with and without associated apoE; extracts from *APOE4* carriers contained more of both types of aggregates than extracts from *APOE3* carriers (Fig. 3B). This finding supports the previous report that *APOE4* carriers have more diffusible Aβ aggregates than *APOE3* carriers[42,43]. Unlike our in vitro aggregations - where the initiation of aggregation is synchronized for all monomers - only ~5% of Aβ aggregates in *APOE4* homozygotes and ~1% of aggregates in *APOE3* homozygotes are co-aggregates (Fig. 3C). This difference probably results from the transience of co-aggregates and the fact that Aβ and apoE are replenished and cleared over years in the CNS. Extracted aggregates therefore

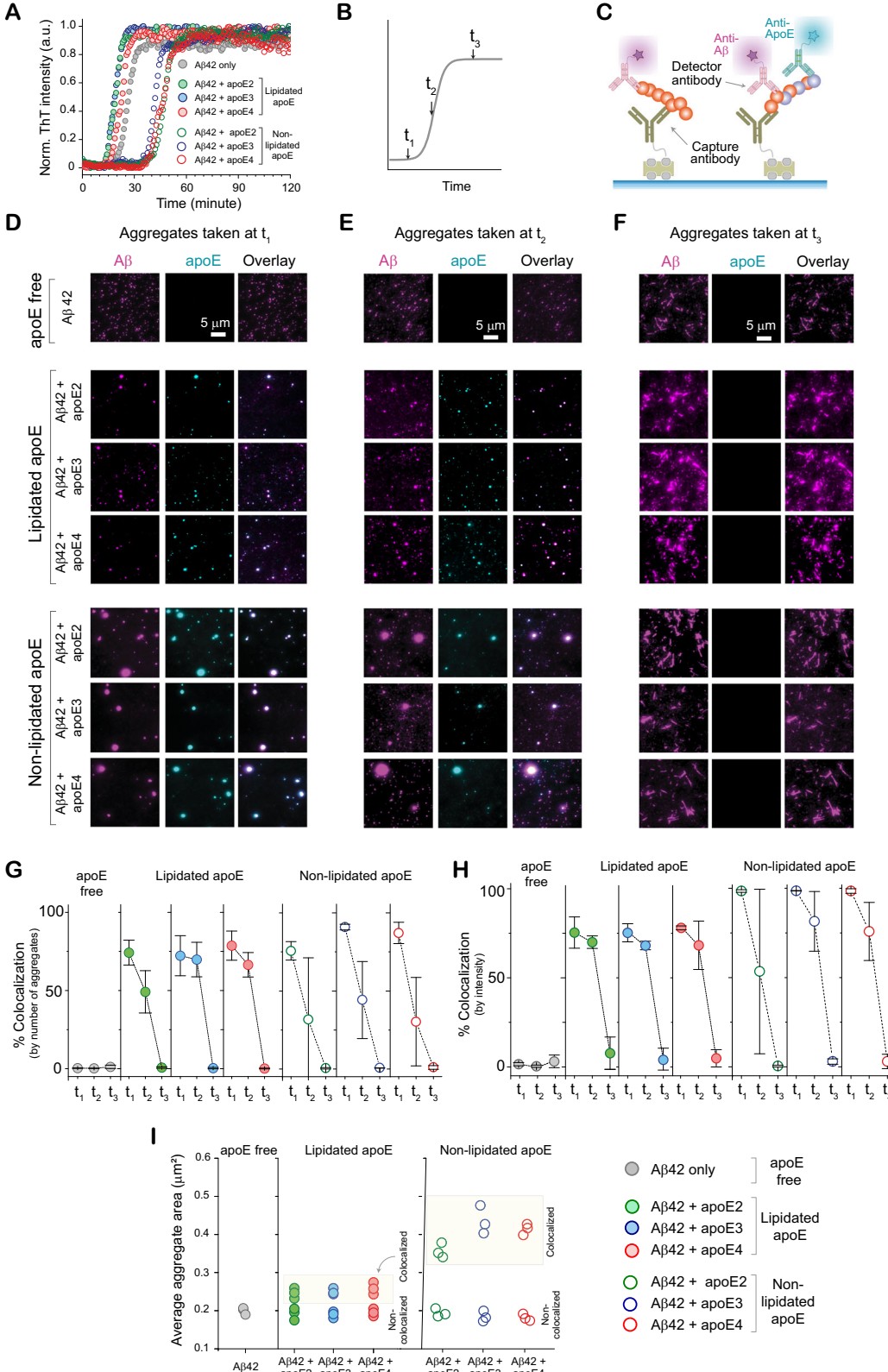

**Fig. 1 | ApoE and Aβ transiently co-aggregate en route to fibrils. A** Aβ42 aggregation (4 μM) in the presence of different non-lipidated isoforms of apoE (0 or 80 nM), monitored by ThT fluorescence (*n* = 3 independent replicates). **B** Time points at which samples were taken for further analysis. **C** SiMPull assay for Aβ42 aggregates and apoE-Aβ co-aggregates (1 μM Aβ monomer equivalents), using biotinylated 6E10 antibody for capture, and Alexa-Fluor-647-labeled 6E10 (500 pM) and Alexa-Fluor-488-labeled EPR19392 (1 nM) antibodies for detection.

**D–F** Representative two-color TIRF images of aggregates were captured at t₁ (**D**), t₂ (**E**), and t₃ (**F**). **G, H** Colocalization between Aβ and apoE at different time points, quantified by aggregate counting (**G**) and 6E10 fluorescence intensity (**H**). Data are plotted as the mean and standard deviation of three independent replicates.
**I** Average sizes of colocalized and non-colocalized aggregates (N.B. in diffraction-limited imaging, the minimum apparent aggregate size is ~0.2 μm²). Source data are provided as a Source Data file.

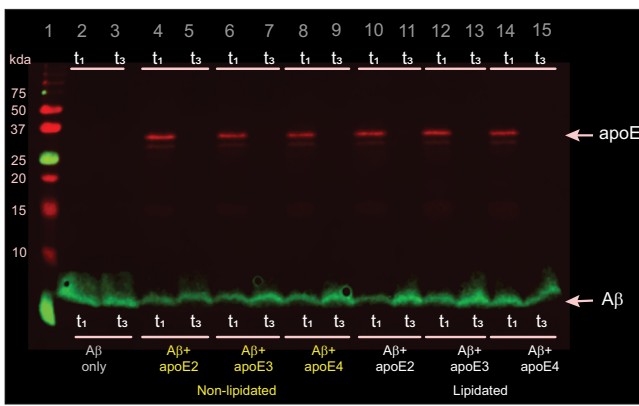

**Fig. 2 | ApoE interacts with Aβ in early-stage aggregates but not fibrils.** Aβ42 aggregates and apoE-Aβ co-aggregates were co-immunoprecipitated using an Aβ-specific antibody (6E10) and then analyzed by western blotting using both Aβ-specific 6E10 and apoE-specific EPR19392 antibodies. This image represent one of three independent experiments.

**Table 1 | Properties of AD postmortem brain tissue samples (Age 58–90)**

| Sex | Postmortem time before freezing (hours) | Braak neurofibrillary tangle stage | *APOE* genotype |
|---|---|---|---|
| F | 24 | 6 | 4/4 |
| M | 69 | 6 | 4/4 |
| F | 10 | 6 | 4/4 |
| F | 70 | 6 | 3/3 |
| F | 26 | 6 | 3/3 |
| F | 57 | 6 | 3/3 |

reflect a broad sample of the reaction pathway, *e.g.* early-stage aggregates alongside fibrils. The low abundance of co-aggregates is consistent with the recent observation that short fibrils are abundant in soaked-brain extracts[44] and our finding that fibrillation leads to shedding of apoE from co-aggregates (Fig. 1). Although few in number, some of the ex vivo co-aggregates are large (Fig. 3E) and contribute disproportionately to the fluorescence intensity. Using the intensity as a proxy for total amounts of protein suggests that co-aggregates comprise 40–60% and 10–35% of the Aβ-aggregate mass in *APOE4* and *APOE3* carriers, respectively (Fig. 3D). These data, therefore, reveal a large isoform dependence in the accumulation of co-aggregates in AD brains.

### Glial Uptake of Early-stage apoE-Aβ Co-aggregates is Isoform- and Lipidation-dependent

Given that all isoforms of apoE, whether lipidated or not, had co-aggregated to similar extents with Aβ42, we next asked whether differential clearance might explain the isoform-dependent accumulation of apoE-Aβ co-aggregates in AD brains. We quantified the uptake of aggregates by two types of cells: i) Human induced pluripotent stem cell (hIPSC)-derived microglia-like cells (iMGLs), and ii) astrocytes generated by reprogramming human fibroblasts *via* induced neuronal progenitor cells iNPCs (iAstrocytes) (Fig. 4A–C, Figure S9). The microglial[45,46] and astrocytic[47,48] identities were confirmed by whole-transcriptome analysis and qRT-PCR previously. Human microglia and astrocytes take up soluble and insoluble Aβ aggregates[49,50], and dysfunction of microglia and astrocytes is implicated in AD[51,52]. We exposed these cells to Aβ aggregates formed at different time points ($t_1$ and $t_3$), in the presence and absence of various lipidated and non-lipidated apoE isoforms. Alongside monitoring aggregate clearance by these cells, we also measured the resulting inflammatory activation by assaying several pro-inflammatory cytokines and one chemokine. Protein aggregates cause neuroinflammation by disrupting CNS homeostasis[53], and neurotoxic Aβ aggregates are known to activate pro-inflammatory pathways in microglia[51] and astrocytes[52].

To isolate the role of co-aggregates, we compared the uptake of early-stage aggregates from $t_1$ (≥75% co-aggregates) and fibrils from $t_3$ (~0% co-aggregates) by iMGLs and iAstrocytes over a 1 h period. Both cell types took up more early stage co-aggregates than fibrils prepared in the presence of lipidated apoE, but not non-lipidated apoE (Fig. 4D, E). Uptake of early-stage co-aggregates by iMGLs was isoform-specific irrespective of lipidation, but only isoform-specific in iAstrocytes if apoE was non-lipidated (Fig. S10A, D). Taken together, these cells internalized early-stage non-lipidated co-aggregates ~2-3-fold

more efficiently if they contained apoE2 rather than apoE4. Co-aggregate clearance is also highly dependent on lipidation status: early-stage lipidated co-aggregates were internalized to a much greater extent than non-lipidated co-aggregates (Figure S11).

To assess the extent of glial activation in response to co-aggregate exposure, we measured the secretion of pro-inflammatory cytokines interleukin-1β (IL-1β,) interleukin-6 (IL-6), tumor necrosis factor-alpha (TNF-α) and chemokine monocyte chemoattractant protein-1 (MCP-1), using enzyme-linked immunosorbent assays (ELISA) (Fig. 5A). Early-stage aggregates ($t_1$) elicited significantly larger release of inflammatory markers than fibrils ($t_3$) if apoE was absent from the aggregation reaction or present in a non-lipidated form. However, the presence of lipidated apoE decreased glial activation by early-stage co-aggregates to levels similar to fibrils (Fig. 5B–I). Cytokine and chemokine secretion was significantly isoform-dependent only for non-lipidated co-aggregates, which promoted secretion in the order apoE4>apoE3>apoE2 (Figure S12). Glial activation was also dependent on the lipidation status of apoE. Co-aggregates induced release of inflammatory markers at higher levels when they were formed in the presence of non-lipidated forms of apoE (Figure S13). ApoE alone, irrespective of isoform or lipidation, inflamed neither iMGLs nor iAstrocytes (Figure S12 I, J).

These responses of iMGLs and iAstrocytes to co-aggregates begin to tease out isoform- and lipidation-dependent patterns that hint at a co-aggregate-mediated link between *APOE* genotype and AD risk. In the presence of non-lipidated apoE, early-stage co-aggregates ($t_1$) induce secretion of pro-inflammatory cytokines and chemokines, but lipidation enhanced their uptake and suppressed inflammation. This pattern suggests a role for early-stage, non-lipidated apoE-Aβ co-aggregates. Within this group, the expected isoform dependence was apparent, *i.e.* non-lipidated apoE4-Aβ co-aggregates were the least well cleared and most inflammatory of all Aβ species.

### Early-stage, Non-lipidated apoE4-Aβ Co-aggregates are toxic and impair clearance by Glial cells

It is widely reported that aggregated Aβ is toxic, and that its toxicity depends on the size, structure, and composition of the aggregate[31,54,55]. To check whether apoE modulates the toxicity of Aβ aggregates in an isoform or lipidation-specific way, we measured the toxicity of Aβ at different stages of aggregation. We assayed this property in two ways: i) the ability to permeabilize lipid membranes and cause Ca$^{2+}$ influx[56,57] and ii) toxicity to the SH-SY5Y human cell line, leading to the release of lactate dehydrogenase (LDH) (Fig. 6A, C). We again compared co-aggregates from $t_1$ to fibrils from $t_3$ - which have shed apoE - in order to isolate the role of co-aggregates. Early-stage co-aggregates induced higher Ca$^{2+}$ influx and LDH release than fibrils when prepared in the presence of non-lipidated apoE, but not lipidated apoE (Fig. 6B, D). Echoing our results on uptake and cytokine/chemokine secretion, the toxicity of co-aggregates was isoform-dependent (apoE4-Aβ being more toxic than apoE2-Aβ) for Aβ aggregates containing non-lipidated apoE, but not lipidated apoE (Figure S14A and S15A). In the absence of Aβ, apoE showed no measurable toxicity in either assay (Figure S14C and Figure S15C). Non-lipidated apoE4-Aβ co-aggregates

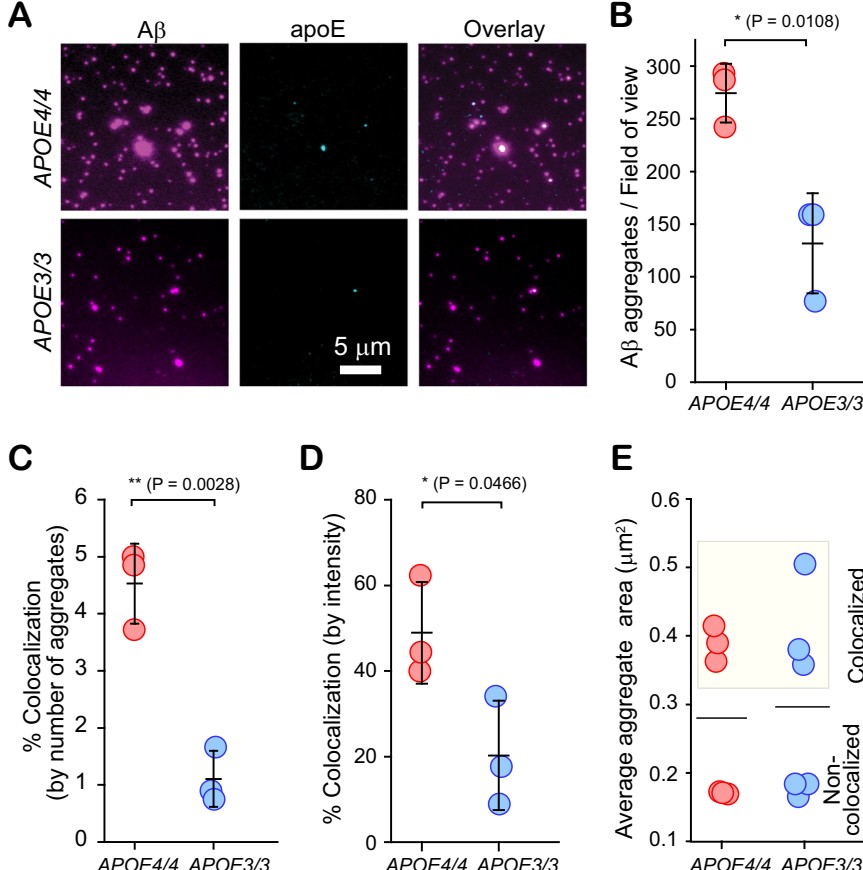

**Fig. 3 | Aβ-apoE co-aggregates form in human brain tissue, but their concentration is isoform-dependent. A** Representative two-color TIRF images of aggregates from frontal-cortex extracts of homozygous *APOE4* and *APOE3* AD patients. **B** Number of Aβ-containing species captured from *APOE4* and *APOE3* homozygotes. **C**, **D** Colocalization between Aβ and apoE in extracts from *APOE4* and *APOE3* homozygotes, quantified by aggregate counting (**C**) and 6E10 (Aβ)

fluorescence intensity (**D**). **E** Sizes of apoE-colocalized and non-colocalized Aβ aggregates in *APOE4* and *APOE3* homozygotes. Data points in panels (**B**–**E**) are plotted as the mean +/− standard deviation of three biological replicates. Statistical significance was calculated using a unpaired two-sample t-test. *$P < 0.05$, **$P < 0.01$. Source data are provided as a Source Data file.

permeabilized lipid membranes more potently (Figure S14B) and induced more LDH secretion (Figure S15B) than the corresponding lipidated forms. This lipidation dependence was absent for fibrils ($t_3$) prepared in the presence of apoE2 and apoE3 isoforms (Figure S15B).

These toxicity assays painted a similar picture: early-stage co-aggregates ($t_1$) were better at permeabilizing lipid bilayers and more toxic to neuroblastoma cells than fibrils, with both lipidation- and isoform-dependent features. Lipidated co-aggregates were less toxic than non-lipidated co-aggregates, and among the latter, toxicity was isoform dependent. Non-lipidated apoE4-Aβ co-aggregates stand out as the most toxic and damaging to lipid membranes.

Having established that non-lipidated apoE-Aβ co-aggregates were especially inflammatory and toxic, we wondered whether their selective removal might rescue Aβ-induced dysfunction. To test this hypothesis, we immunoprecipitated early-stage non-lipidated apoE4-Aβ co-aggregates from a 1:1 mixture of lipidated and non-lipidated co-aggregates using HAE-4, an antibody specific for non-lipidated apoE4[16,24,27], and an isotype control (Fig. 7A, B). Cellular uptake and neuroinflammation assays on the mixture and supernatant showed that removing non-lipidated co-aggregates did indeed enhance Aβ clearance and reduced Aβ-mediated secretion of inflammatory markers (MCP-1, Il-1β, TNF-α, IL-6) in both iMGLs (Fig. 7C–G) and iAstrocytes (Fig. 7H–L). Immunoprecipitating non-lipidated apoE4-Aβ co-aggregates also reduced membrane damage (Fig. 7M) and LDH release (Fig. 7N). It is counterintuitive that removing the non-lipidated half of the aggregates increased the total uptake of Aβ (by

~50% for iAstrocytes and ~100% for iMGLs), even though there is ~50% less total Aβ. However, exclusively lipidated apoE4-Aβ co-aggregates (Fig. 4D–E) were taken up at well over double the levels of a 1:1 mixture of lipidated and non-lipidated co-aggregates. This implies that the inflammation caused by contact with non-lipidated co-aggregates impairs the ability of iAstrocytes and iMGLs to clear lipidated aggregates. Sparing cells from this insult by removing non-lipidated co-aggregates therefore increases the clearance of lipidated co-aggregates.

**Early-stage apoE4-Aβ Co-aggregates detected in AD brains are poorly lipidated**

Finally, to corroborate the putative role of non-lipidated apoE co-aggregates, we asked what proportion of co-aggregates contain non-lipidated apoE. We were unable to source a specific antibody to either lipidated or non-lipidated apoE3 with sufficiently low non-specific or cross binding, so these experiments were conducted solely on samples from *APOE4/4* homozygotes AD patients. Immunoprecipitating non-lipidated apoE from these brain extracts using the HAE-4 antibody (Fig. 8A) disproportionately removed the largest co-aggregates (Fig. 8B). The drop in colocalization after immunoprecipitation (Fig. 8C, D) suggests that the vast majority of co-aggregates (~80%) trapped in the neural tissue of AD patients contain sufficient non-lipidated apoE to be pulled down, despite the fact that only a minority of apoE in the CNS is non-lipidated. However, it is important to note that because co-aggregates can contain both lipidated and non-

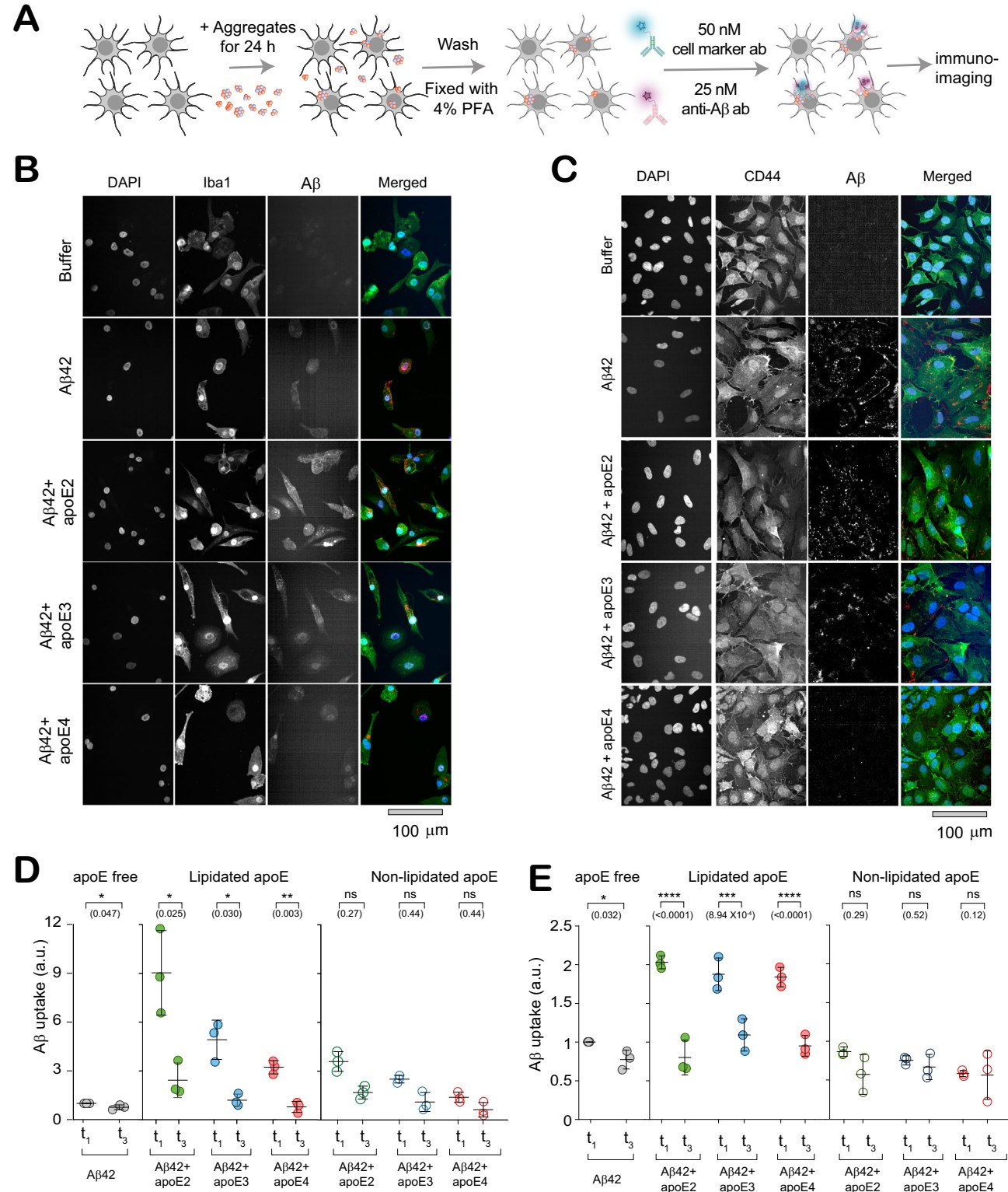

**Fig. 4 | Clearance of apoE-Aβ co-aggregates by glial cells depends on aggregate maturity, apoE isoform and lipidation. A** Assay for uptake of aggregates by glial cells. **B**, **C** Representative two-color epifluorescence images showing uptake of Aβ42 (1 µM monomer equivalents) and non-lipidated apoE (0 or 80 nM) from aggregation mixtures at $t_1$ (end of lag phase) by iMGLs (**B**) and iAstrocytes (**C**). **D**, **E** Quantified uptake of Aβ by iMGLs (**D**) and iAstrocytes (**E**). Units of uptake = integrated fluorescence of sample divided by the integrated fluorescence of internalized early-stage Aβ42 aggregates at $t_1$ for each replicate. Data points are plotted as the mean +/− standard deviation of three biological replicates. Statistical significance was calculated using a unpaired two-sample $t$-test. *$P < 0.05$, **$P < 0.01$, ***$P < 0.001$, ns, non-significant ($P \geq 0.05$). Source data are provided as a Source Data file.

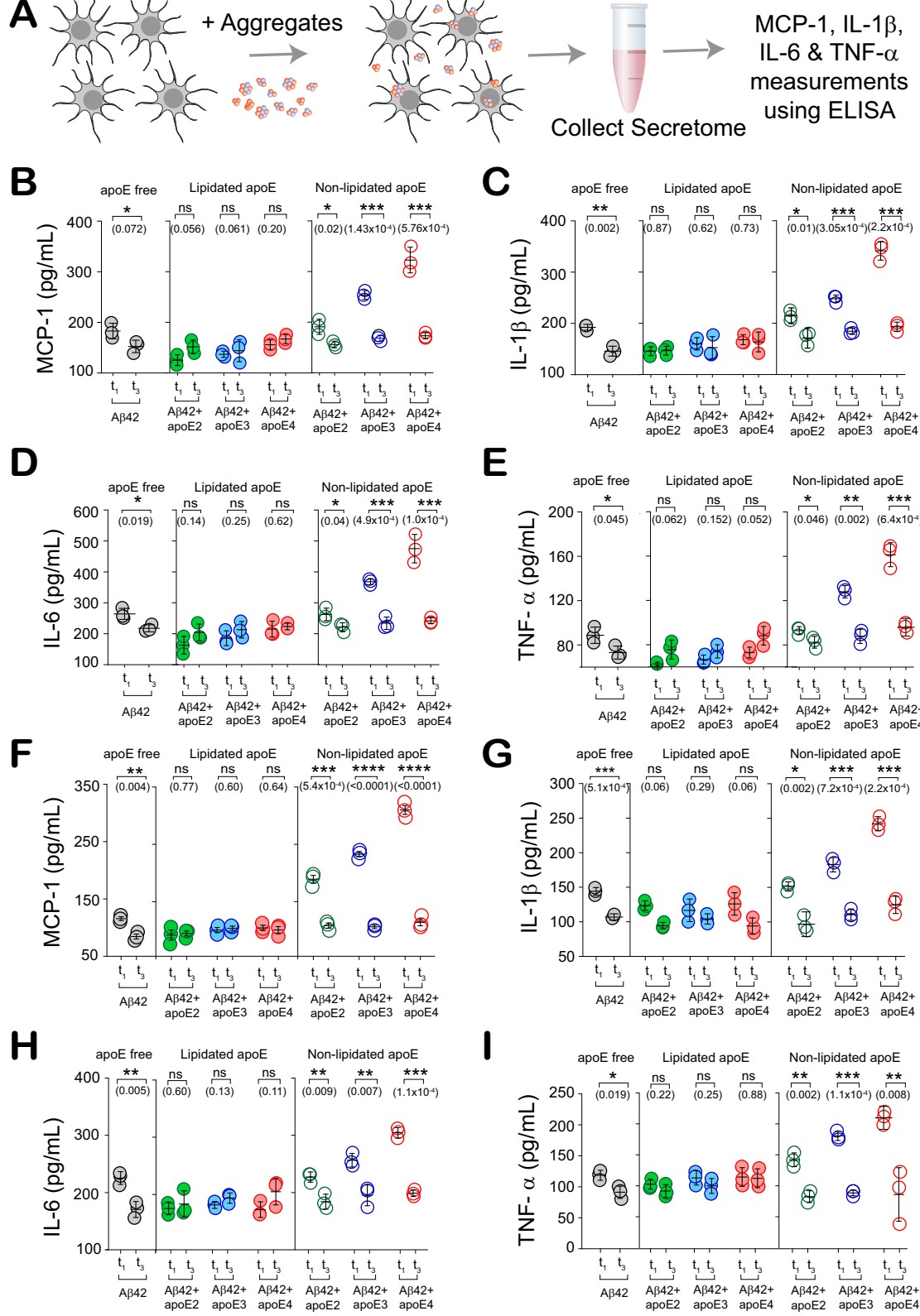

**Fig. 5 | Early-stage, non-lipidated apoE-Aβ co-aggregates inflame glial cells in an isoform-dependent way. A** Aggregate-induced cytokine and chemokine secretion by iMGLs and iAstrocytes. **B**–**I** MCP-1, IL-1β, IL-6 and TNF-α release by iMGLs (**B**–**E**) and iAstrocytes (**F**–**I**) induced by early-stage (t₁) and fibrillar (t₃) aggregates. Each data point represents one of three biological replicates; error bars represent mean values +/− standard deviation. Statistical significance was calculated using a unpaired two-sample t-test. *P < 0.05, **P < 0.01, ***P < 0.001, ns, non-significant (P ≥ 0.05). Source data are provided as a Source Data file.

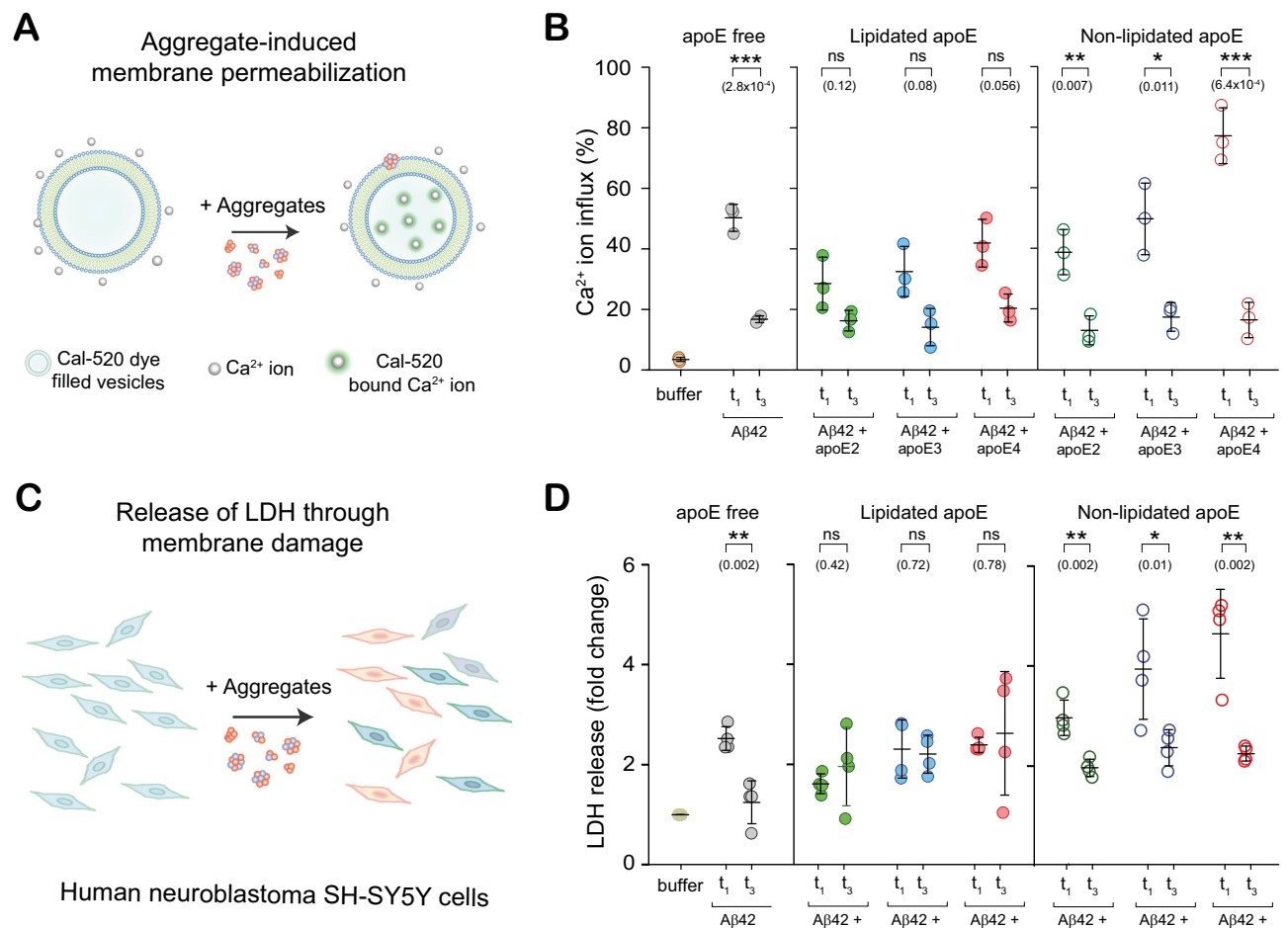

**Fig. 6 | ApoE modulates the toxicity of Aβ aggregates isoform-dependently.**
**A**, **B** Permeabilization of lipid bilayers by early-stage ($t_1$) and fibrillar ($t_3$) Aβ aggregates ([Aβ42] = 4 μM in monomer equivalents; [apoE] = 0 or 80 nM). $Ca^{2+}$ influx is referenced to the influx caused by the ionophore, ionomycin.
**C**, **D** Neurotoxicity of $t_1$ and $t_3$ aggregates to human neuroblastoma SH-SY5Y cells, assayed by lactate dehydrogenase (LDH) release ([Aβ42] = 4 μM in monomer

equivalents; [apoE] = 0 or 80 nM). Data points represent one of three independent experiments (**B**) or one of three or four biological replicates (**D**); error bars represent mean values +/− standard deviation of three independent (**C**) or four biological (**D**) replicates. Statistical significance was calculated using a unpaired two-sample t-test. *$P < 0.05$, **$P < 0.01$, ***$P < 0.001$, ns, non-significant ($P \geq 0.05$). Source data are provided as a Source Data file.

lipidated apoE molecules, this does not imply that 80% of apoE in co-aggregates is fully non-lipidated.

## Discussion

It is widely thought that Aβ species formed in the early stages of aggregation are key molecular players in AD: they trigger neuronal dysfunction, impair synaptic plasticity and their abundance correlates with severity of neurodegeneration in AD[58,59]. Attempts to reconcile this hypothesis with the link between *APOE* inheritance and late-onset AD risk have so far been unsuccessful. This study identifies a potential role for direct interactions between apoE and early-stage Aβ aggregates in forging that link. Starting from the premise that apoE could affect AD risk by interacting with Aβ aggregates that are either rare or transient, we sampled a broad cross-section of the aggregation equilibrium. We found that apoE-Aβ co-aggregates are transient but not rare: the journey to apoE-free fibrils passed through highly populated apoE-containing intermediates in vitro, irrespective of isoform or lipidation of apoE. By contrast, there were stark isoform- and lipidation-dependent differences in the abundances of ex vivo co-aggregates. Co-aggregates comprise more of the Aβ mass extracted from the frontal cortices of *APOE4/4* patients (~50%) than *APOE3/3* patients (~20%). About 80% of ex vivo co-aggregates detected from

*APOE4* homozygotes contain at least some of the less-common, non-lipidated form of apoE.

Previous studies have identified apoE in Aβ deposits in the human brain as well as in plasma and cerebrospinal fluid (CSF)[60–62]. The extent of apoE-Aβ complex formation remains a matter of debate, with some studies suggesting minimal interaction in vitro and only around 5% of apoE-Aβ complexes in human cerebrospinal fluid[28]. The challenge of detecting and quantifying these complexes, coupled with the variability introduced by apoE isoform, lipidation state, and Aβ aggregation state has made it difficult to resolve this question. This work offers a fresh perspective by studying the relevant variables using single-molecule imaging methods. We found that although only 1–5% of ex vivo Aβ particles are associated with apoE, these co-aggregates account for ~20–50% of total Aβ. This finding, along with the transience of co-aggregates on the aggregation pathway, may explain why previous studies have produced such divergent results. Beside, the observed connection between apoE and Aβ within plaques of AD patients[24,63], despite apoE not directly binding to fibrillar Aβ, can be attributed to various factors. These include shared interactions of apoE and Aβ with receptors on glial cells such as LDLR and LRP1[6], the involvement of bridging molecules like heparin that link apoE and Aβ[64], and the natural inclination of

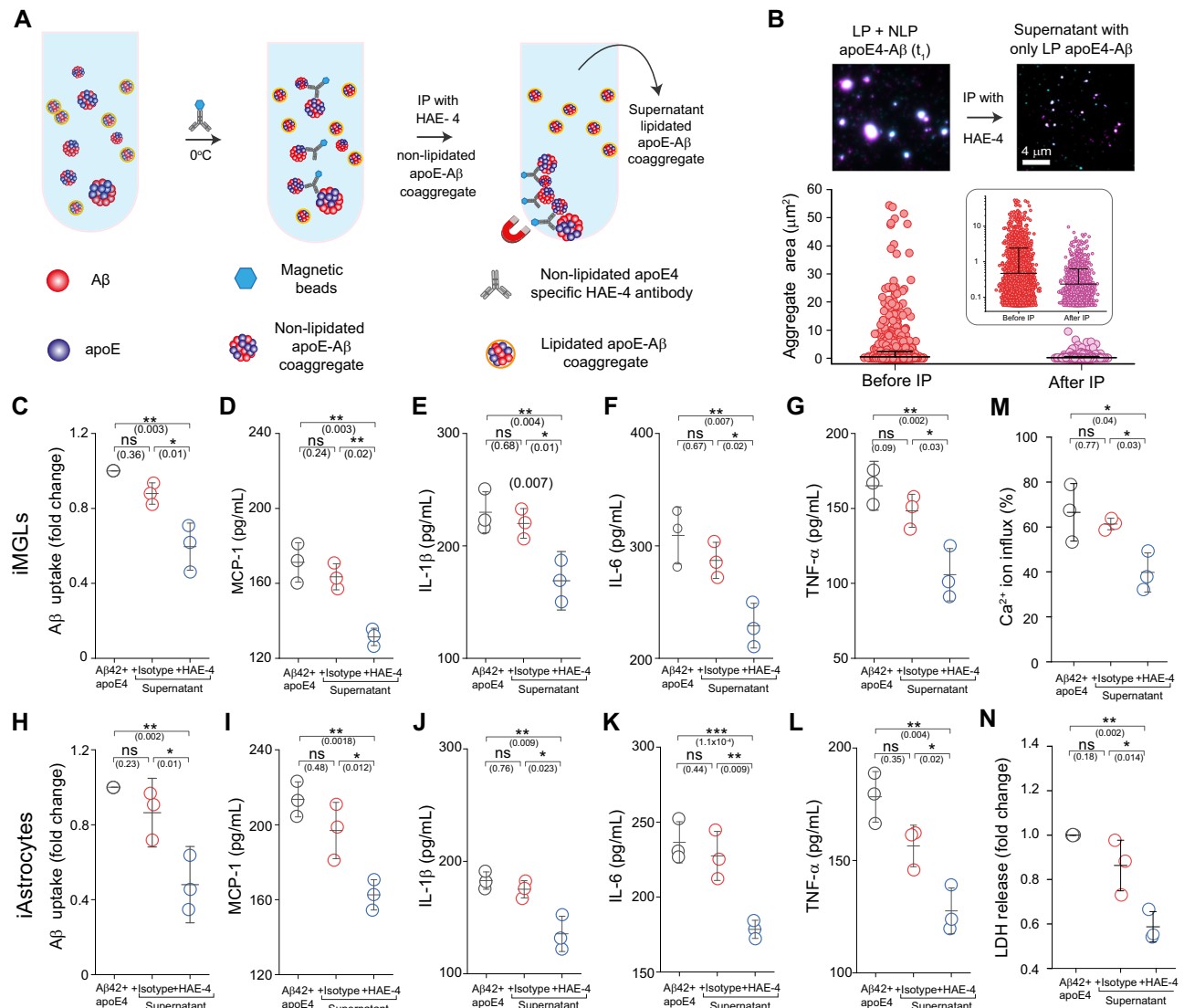

**Fig. 7 | Removing non-lipidated apoE4-Aβ co-aggregates enhances Aβ clearance, reducing secretion of inflammatory markers and cytotoxicity.**
**A** Immunoprecipitation of non-lipidated early-stage co-aggregates from a mixture of non-lipidated and lipidated apoE4-Aβ co-aggregates using the HAE-4 antibody. **B** Representative two-color TIRF images and sizes of individual apoE4-Aβ co-aggregates before (number of aggregates: 10,277, average size: 0.47 ± 1.95 μm²) and after (number of aggregates: 4354, average size: 0.23 ± 0.39 μm²) immunoprecipitation (data are plotted in log₁₀ scale in inset). **C–L** Effect of immunoprecipitation on Aβ uptake and cytokine/chemokine release by iMGLs (**C–G**) and iAstrocytes (**H–L**). **M, N** Effect of immunoprecipitation on neurotoxicity of apoE4-Aβ co-

aggregates, measured by permeabilization of lipid membranes (**M**) and LDH release by human neuroblastoma SH-SY5Y cells (**N**). Data points represent one of three biological replicates (**C–L, N**) or one of three independent experiments (**M**); Units of uptake = integrated fluorescence of sample divided by the integrated fluorescence of internalized early-stage Aβ42 aggregates at t₁ for each replicate, as in Fig. 3 (**C, H**); error bars represent mean values +/− standard deviation of three biological replicates. Statistical significance was calculated using one-way ANOVA with post-hoc Tukey. *$P < 0.05$, **$P < 0.01$, ***$P < 0.001$, ns, non-significant ($P \geq 0.05$). Source data are provided as a Source Data file.

both apoE and Aβ to aggregate in acidic environments[6] such as lysosomes, all of which may significantly contribute to their combined presence in plaques.

The lack of consensus on the nature of apoE-Aβ interactions has prompted many hypotheses for how *APOE* genotype influences AD risk, some complementary and some contradictory[5,6,26]. In particular, there are two major mechanistic explanations for the undisputed observation that apoE4 increases Aβ deposition: either apoE4 causes more Aβ aggregates to form, or inhibits their clearance. Our results support the latter explanation. We found that apoE isoforms do not affect the kinetics and thermodynamics of aggregation, but strongly influence cellular uptake and toxicity. In the CNS, a significant amount of Aβ aggregates are removed for degradation by glial cells via phagocytosis, pinocytosis, or receptor-mediated endocytosis[49,65].

Exposure to aggregates stresses iMGLs and iAstrocytes, which then release pro-inflammatory cytokines such as IL-1β, IL-6, TNF-α and chemokine MCP-1, damaging nearby neurons and promoting neurodegeneration[66,67]. We found that co-aggregation with lipidated apoE enhanced Aβ uptake, while non-lipidated apoE4-Aβ co-aggregates were the slowest cleared and most inflammatory of all the aggregates that we studied. Our results suggest that apoE interacts with newly formed aggregates in the extracellular space; if the co-aggregate contains lipidated apoE, this accelerates its clearance and avoids Aβ-induced toxicity. On the other hand, co-aggregates containing non-lipidated apoE4 are poorly cleared and induce a proinflammatory state in glial cells, which impairs the clearance of other Aβ aggregates (Fig. 9). This dynamic would lead to greater deposition of Aβ over time in individuals with *APOE4* genotypes. The fact that apoE4

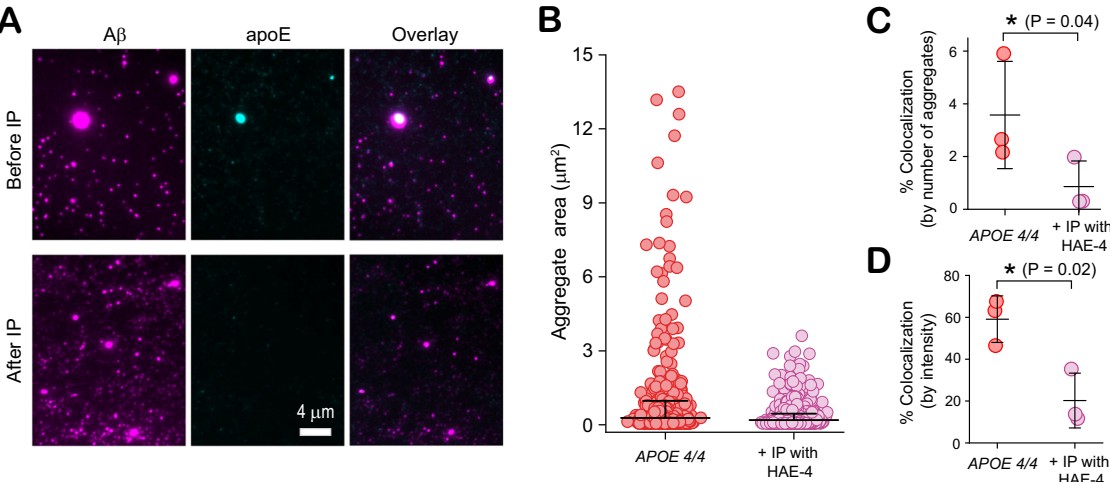

**Fig. 8 | Early-stage apoE4-Aβ coaggregates from the brains of *APOE4/4* AD patients are poorly lipidated. A** Representative two-color TIRF images of early-stage aggregates extracted from frontal cortices of *APOE4* homozygotes, before and after immunoprecipitation (IP) with the non-lipidated-apoE4-specific antibody, HAE-4. **B** Sizes of individual Aβ-containing species before (number of aggregates: 3814, average size: $0.28 \pm 0.69$ μm) and after (number of aggregates: 3576, average size: $0.19 \pm 0.26$ μm) immunoprecipitation (data are plotted in $\log_{10}$ scale in inset).

**C, D** Effect of immunoprecipitation on colocalization between Aβ and apoE4 quantified by aggregate counting (**C**), and fluorescence intensity (**D**). Data points in panels (**C, D**) represent one of three biological replicates; error bars represent mean values +/− standard deviation of theree independent experiments. Statistical significance was calculated using a paired two-sample t-test. *$P < 0.05$, **$P < 0.01$, ***$P < 0.001$, ns, non-significant ($P \geq 0.05$). Source data are provided as a Source Data file.

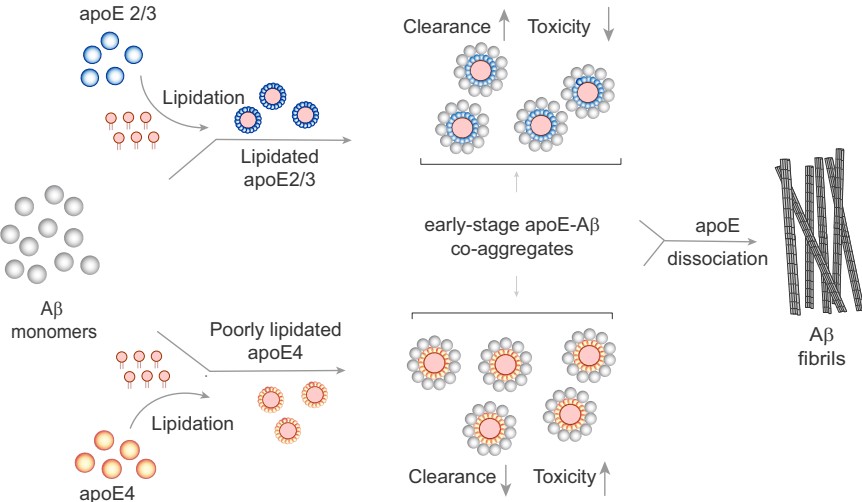

**Fig. 9** | Proposed mechanism for how apoE influences Aβ aggregation and modulates Aβ clearance and toxicity in an isoform- and lipidation-specific manner.

is less lipidated in vivo[68,69] than apoE2 or apoE3 may exacerbate this vicious circle.

This apoE-mediated mechanism for Aβ deposition, combined with co-aggregate-induced inflammation and toxicity may influence disease progression in AD. Non-lipidated apoE-Aβ co-aggregates are more likely to be deposited than lipidated co-aggregates, but are also more toxic. This may explain why deleting the ATP-binding cassette transporter 1 (ABCA1) - which regulates apoE lipidation in the CNS[17] – increases the deposition of Aβ plaques[70,71], while overexpressing or upregulating ABCA1 relieves cognitive impairment caused by Aβ and reverses memory deficits in transgenic animals[72]. Our findings connect the *APOE4* genotype with increased amyloid load, as well as faster AD onset and progression, via a possible culprit: non-lipidated apoE4-Aβ co-aggregates formed in the early stages of aggregation. Importantly, mature Aβ fibrils do not contain apoE and are functionally indistinguishable, whichever form of apoE was present during aggregation. This finding strengthens our hypothesis that apoE contributes to AD risk by modulating the behaviour of early-stage Aβ aggregates.

Our work aims to address a significant gap in understanding by investigating how Aβ42 interacts with various isoforms of apoE in both lipidated and non-lipidated states, and by assessing how these interactions affect disease-relevant functions of Aβ42. We observed a stark contrast in the interactions between apoE and Aβ across the early (t_1) and later (t_3) stages of Aβ aggregation. Isoform-specific and lipidation-dependent differences in Aβ uptake and inflammation observed in early-stages (t_1) converged into a uniform response in later stages (t_3), indicating stage-specific interactions between Aβ and apoE. Furthermore, we observed functional differences when Aβ was associated with lipidated versus non-lipidated apoE. At the early stages of Aβ aggregation (t_1), glia-mediated clearance of Aβ was largely influenced by lipidated apoE, while the secretion of proinflammatory markers was predominantly induced by non-lipidated apoE. These findings highlight the critical influence of apoE lipidation at the early stages of Aβ aggregation. However, as the aggregation progresses to t_3, the presence of apoE within the Aβ aggregates diminishes, along with its impact on Aβ function. Although there are slight variations in

experimental trends, our study reflects the importance of considering the temporal aspects of apoE's influence on Aβ aggregation, clearance, and inflammation, which are crucial for understanding the mechanisms underlying AD pathogenesis.

Although we hope this study will go some way to settling the controversy around apoE's role in AD, it is important to acknowledge its limitations. Firstly, our study does not rule out contributions from Aβ-independent mechanisms, such as apoE's influence on neuronal susceptibility to injury, insulin signaling, or neuroinflammation[1,6]. Similarly, we did not study Aβ clearance by mechanisms other than cellular uptake, such as enzymatic degradation, or transport across the blood-brain barrier or into the CSF. Secondly, we currently have limited structural information on apoE-Aβ co-aggregates. Analyzing the composition and high-resolution structure of co-aggregates isolated over the course of AD is likely to shed further light on apoE's influence on Aβ. Our hypothesis suggests that structural or stoichiometric differences between co-aggregates affect receptor interactions, an area that warrants further investigation. Third, deciphering a detailed kinetic model of Aβ aggregation in the presence of apoE is likely to be important to understanding this fundamental process of AD. This is particularly important in light of our finding that Aβ aggregation passes through highly populated co-aggregate intermediates if apoE is present. These combined efforts will contribute to a more comprehensive understanding of apoE's role in AD pathogenesis.

Our work suggests further avenues for investigating and treating late-onset AD. We show that removing non-lipidated apoE4 co-aggregates reduces secretion of pro-inflammatory markers, enhances Aβ uptake by glia cells, and protects from Aβ-induced membrane disruption and subsequent Ca[2+] dysregulation. This may form the molecular basis of immunotherapy with the HAE-4 monoclonal antibody raised against non-lipidated human apoE[16,24]. We found opposing trends in the isoform-dependence of internalization and membrane permeabilization of co-aggregates, suggesting that uptake may be mainly receptor-mediated. Because the receptor binding capacity of apoE depends on its lipidation[73,74], the fact that lipidation of co-aggregates enhances their uptake supports this idea. ApoE's influence on endocytosis is well studied, with roles for other AD risk factors such as LDL[75], TREM2[76] and *PICALM*[77], while LRP1 and VLDLR are putative receptors for the transport of apoE-Aβ complexes[29]. Validating specific receptor(s) which play a significant role in the internalization of large co-aggregates could present opportunities to precisely target apoE in AD. Modulating the formation or uptake of non-lipidated, early-stage co-aggregates in an isoform-specific way could eliminate the AD-specific role of apoE without impeding its essential functions in lipid transport. Enhancing apoE lipidation could also provide a new intervention strategy for late-onset Alzheimer's disease, specifically for *APOE4* carriers, because apoE4 is less lipidated than apoE2 or apoE3[6,69].

## Methods

### Ethics statement
Ethical approval for the the post-mortem tissue was granted through the Newcastle and North Tyneside Research Ethics Committee. Data relating to supplied tissue were released on an anonymized basis. For iMGLS and iAstrocytes, human skin fibroblast samples were obtained for a different study under the Yorkshire and Humber Research and Ethics Committee number: 16/YH/0155.

### Aggregation of recombinant Aβ42
The monomeric recombinant Aβ42 peptide (purchased from Stratech, Cat. No. A-1167-2, Manufacturer: rPeptide) was prepared by resuspending it in a solution of 1% NH₄OH in phosphate-buffered saline (PBS) buffer at a concentration of 1 mg/mL, following the manufacturer's instructions. To remove any insoluble components, the mixture was centrifuged at -4000 g for 30 s. Subsequently, the Aβ42 concentration of the solution was determined using a nanodrop

spectrophotometer. Next, the peptide solution was diluted into PBS at a concentration of 200 μM and 50 μL aliquots prepared on ice. The aliquots were then flash frozen on liquid nitrogen and stored at −80 °C for future use. Subsequently, these aliquots were diluted in PBS to a concentration of 4 μM and aggregated in a 96-well half-area plate (Corning, Cat. No. 3881) at 37 °C without shaking in the absence or presence of non-lipidated or lipidated apolipoprotein E (apoE) isoforms E2, E3 or E4 (Peprotech, Cat. No. 350-12, 350-02 or 350-04) at concentrations of 80 nM (lipidated apoE2, apoE3 and apoE4, and non-lipidated apoE2 and apoE3) or 20, 40 and 80 nM (non-lipidated apoE4). The aggregation process was monitored using 20 μM Thioflavin T dye (Sigma, Cat. No. T3516) using a plate reader (Clariostar Plus, BMG Biotech). At concentrations below 10 μM, apoE forms tetramers in the absence of lipids. However, in the presence of lipids apoE form large complexes. To be consistent, we have reported the apoE concentration based on its monomeric concentration.

To analyze the species formed during Aβ42 aggregation and co-aggregation with apoE, samples of the aggregation mixture (aggregated without ThT) were taken at the end of the lag phase ($t_1$), -15 minutes' incubation (30 min and 10 min for co-aggregation with non-lipidated and lipidated apoE respectively), middle of the growth phase ($t_2$) - 30 minutes' incubation (45 min and 20 min for co-aggregation with non-lipidated and lipidated apoE respectively) and at the plateau phase ($t_3$) - 90 minutes' incubation (120 min for co-aggregation with both non-lipidated and lipidated apoE). For each condition, samples at timepoints $t_1$, $t_2$, and $t_3$, were collected from different wells, to minimize any potential confounding effects arising from alterations in volume, mixing, or mechanical instability during sample collection.

### Lipidation of recombinant ApoE
ApoE was lipidated following a previously published protocol[28]. All the isoforms of apoE were lipidated using a cholate dialysis method using 1-palmitoyl-2-oleoyl-glycerol-3-phosphocholine 16:0-18:1 PC (POPC) (Avanti Lipids, 850457 C) and Cholesterol (Avanti Lipids, 700100 P) in a molar ratio 1:90:5 (apoE: POCP: cholesterol). These ratios of apoE and lipids were selected to mimic the physiological lipid composition of HDL-like apoE particles. Lipids can be directly incorporated into recombinant apoE, but the sodium cholate dialysis method has been shown to produce lipidated apoE homogeneously and reproducibly[28,78].

First, 166.67 μg recombinant apoE was suspended in 600 μL of apoE buffer [20 mM phosphate buffer (Sigma) containing 50 mM NaCl (Sigma, Cat. No. S7653), 1 mM dithiothreitol (DTT, Sigma, Cat. No. 10708984001), and 1 mM ethylenediaminetetraacetic acid (EDTA, Sigma, Cat. No. E9884) at pH 7.4]. In parallel, stock solutions of 10 mg/mL 16:0-18:1 PC (Avanti Lipids, Cat. No. 850457 C), and 10 mg/mL cholesterol (Avanti Lipids, Cat. No. 700100 P) in chloroform were made. Then 16:0-18:1 PC (18.3 μL) and cholesterol (2 μL) were mixed in a glass vial at a molar ratio of 90:5 and the chloroform was then removed under vacuum in a desiccator overnight. Then 40 μL of Dulbecco's phosphate-buffered saline buffer (DPBS, ThermoFisher, Cat. No. 141901440) with 0.5 mM DTT (Sigma) was added to each vial to resuspend the lipid at a total lipid concentration of 5 μg/μL, which was then vortexed three times for 10 min at 2 min intervals. Then the solution was left to stand for 30 min at room temperature. Then, sodium cholate (50 mg·mL⁻¹, Sigma) was slowly added to the lipid solutions with vortexing until the solution was free of turbidity. The apoE solution was then added and the mixture vortexed three times for 10 min, at 5 min intervals and left to stand for 1 h. Then the entire solution was dialyzed using Slide-A-Lyzer Dialysis Cassettes with a 10 kDa -cut off (ThermoFisher Cat 66380) with apoE buffer for 72 h, (with a buffer change every 24 h) at 4 °C. After this, the apoE solutions were purified by gel filtration chromatography (Superose 6 Increase 3.2/300) at 4 °C, and the fractions containing lipidated apoE were

combined and concentrated using an Amicon Ultra 15 mL 10 kDa cut-off centrifugal filter unit (Merck Millipore, Cat. No. UFC901024) at 3000 g for 20 min at 4 °C. The concentration of lipidated apoE was determined by absorbance measurements at 280 nm using an extinction coefficient of 44,460 $M^{-1}\cdot cm^{-1}$. Finally, the samples of lipidated apoE were stored at 4 °C. Lipidation of all the apolipoproteins were performed in parallel using the same lipid–cholesterol suspension.

## Postmortem brain tissue

Flash-frozen brain tissues from six AD patients (three homozygous *APOE4* and three homozygous *APOE3* AD patients) were obtained from the Newcastle brain tissue resource (NTBR) under a Material Transfer Agreement. Tissue was processed at the NTBR, where regions of interest were removed from the frontal cortex (Brodmann area 9) and immediately frozen at −80 °C.

## Extraction of diffusible protein aggregates from postmortem tissue

Diffusible Aβ-containing aggregates were extracted from human tissue following a recently published protocol with minor changes[40]. In summary, the supplied tissues were chopped into 300 mg pieces using a razor blade and incubated with 1 mL of artificial cerebrospinal fluid (aCSF) buffer (120 mM NaCl, 2.5 mM KCl, 1.5 mM $NaH_2PO_4$, 26 mM $NaHCO_3$, 1.3 mM $MgCl_2$, pH 7.4) at 4 °C for 30 min under gentle agitation. The mixture was then centrifuged for 10 min at 2000 g at 4 °C. The upper ~80% of the supernatant was further centrifuged for 110 min at 14,000 g at 4 °C. The upper ~80% of the supernatant was collected and dialyzed using a 2 kDa molecular-weight-cut-off (MWCO) Slide-A-Lyzer (ThermoFisher) for 72 h at 4 °C, against a 50-fold excess of aCSF buffer, which was changed every 12 h. These samples were aliquoted into small volumes and stored at −80 °C, and used for further experiments with no further freeze-thaw cycles.

## Materials for SiMPull assays

Biotinylated 6E10 (Biolegend, Cat. No. 803007, Lot No. B230416), Alexa-Fluor-647-labeled 6E10 (Biolegend, Cat. No. 803021, Lot No. B304121), Alexa-Fluor-488-labeled anti-apoE antibody F-9 (Santa Cruz Biology, Cat. No. sc-390925 AF488, Lot No. B1717), Alexa-Fluor-647-labeled Mouse IgG1 (isotype control, MGL, Cat. No. M075-A64, Lot. No. 002), Alexa-Fluor-488-labeled Mouse IgG (isotype control, Fisher Scientific, Cat. No. 65-0865-14) antibodies were used without further modification or purification and stored at 4 °C until needed. Anti-apoE antibody EPR19392 (Abcam, Cat. No. ab227993, Lot No. GR3268327-5,130 μg, at 1.3 mg/mL in PBS) was mixed with twenty molar equivalents of Alexa Fluor 488 TFP ester (17.2 nmol, 8 mM in dimethyl sulfoxide) and 10 μL of 1 M $NaHCO_3$, and incubated for 2 h. The excess dye was removed using a Zeba Spin Desalting Column (7 kDa MWCO), then an Amicon spin filter (50 kDa MWCO), and the concentration of antibody and degree of labeling ( ~ 1.5 dyes per antibody) were measured using A495 and A280 (Nanodrop, ThermoFisher). The labeled antibody was diluted to 0.5 mg/mL in 0.5 x PBS containing 0.01% sodium azide and 50% glycerol and stored at −20 °C until needed. NeutrAvidin (ThermoFisher, Cat. No. 31000) and bovine serum albumin (BSA, ThermoFisher, Cat. No. 10829410) were used without further purification and stored at 4 °C and −20 °C, respectively. Glass coverslips (26 × 76 mm, thickness #1.5, VWR, Cat. No. MENZBC026076AC40) were covalently PEGylated and stored in a desiccator at −20 °C until needed.

## SiMPull protocol

Assay wells were first coated with 10 μL of 0.2 mg/mL NeutrAvidin in 1× PBS containing 0.05% Tween-20 for 5 min, washed twice with 10 μL of 1× PBS containing 0.05% Tween-20, and once with 10 μL of 1× PBS containing 1% Tween-20. Ten microliters of biotinylated 6E10 (10 nM in 1× PBS containing 0.1 mg/mL BSA) were added to each well for 15 min. The wells were then washed twice with 10 μL of 1× PBS containing

0.05% Tween-20, and once with 10 μL of 1× PBS containing 1% Tween-20. The (co-)aggregates taken at different time points (diluted to 1 μM monomer equivalents in PBS) or soaked-brain extracts (neat) were then added and incubated for 1 h. The wells were then washed twice with 10 μL of 1× PBS containing 0.05% Tween-20, and once with 10 μL of 1× PBS containing 1% Tween-20. The mixture of imaging antibodies (1 nM Alexa-Fluor-488-labeled EPR19392 and 500 pM Alexa-Fluor-647-labeled 6E10 in 1× PBS containing 0.1 mg/mL BSA) were then added and incubated for 30 min. The wells were then washed twice with 10 μL of 1× PBS containing 0.05% Tween-20, and once with 10 μL of 1× PBS containing 1% Tween-20. Finally, 3 μL of 1× PBS were added to each well and samples were sealed using a second plasma-cleaned coverslip.

## Imaging setup for SiMPull

Samples were imaged using a custom-built total internal reflection fluorescence (TIRF) microscope based on a Nikon Ti2 microscope fitted with a Perfect Focus unit. Laser beams (488 nm and 635 nm, Oxxius) were circularly polarized using a quarter-wave plate (WPQ05M-405, Thorlabs), then focused onto the back focal plane of a 100x Plan Apo TIRF, NA 1.49 oil-immersion objective lens (Nikon). Fluorescence emission was collected by the same objective and separated from the excitation beam using a dichroic beamsplitter (Di01-R405/488/561/635, Laser 2000). The emitted light was passed through a 488 nm long-pass filter (BLP01-488R, Laser 2000) and bandpass filter (FF01-520/44-25, Laser 2000) for Alexa-Fluor-488 emission, and a 635 nm long-pass filter (BLP01-635R-25, Laser 2000) for Alexa-Fluor-647 emission and imaged on an EMCCD camera (Photometrics Evolve). The open-source software Micro-Manager 1.4 was used to automate image acquisition[79]. Images were acquired in an unbiased way at laser powers of ~10 W/cm² and averaged over 50 frames with an exposure time of 50 ms each.

## Analysis of SiMPull data

The averaged images acquired with excitation at 488 nm (apolipoprotein E detection channel) and 635 nm (amyloid-beta detection channel) were passed to the Fiji plugin ComDet3[80] with a self-written Python automation code. Particles are differentiated from noise by selecting 20 times intensity significance (20x SD) compared to the background. For spots in two different channels to be considered colocalized, the displacement between their centers of mass (determined by Gaussian fitting) was required to be ≤3 pixels. Colocalization by aggregate number was defined as the ratio between the number of colocalized spots and the total number of spots in the Aβ channel. Colocalization by intensity was defined as the ratio between the sum of the integrated intensities of colocalized spots and all spots in the amyloid-beta channel. Random colocalization was estimated by performing the colocalization algorithm with one of the images (50 μm x 50 μm) rotated by 90 degrees, which was less than 1% for all data sets.

## Western blot analysis of pure apoE and Aβ42

Pure samples of apoE2, apoE3, apoE4 (2 μg each), and Aβ42 peptide (100 ng) were resolved on 16.5% Tris-Tricine SDS gels (Bio-Rad 4563063) using 1x Tris-Tricine running buffer (Bio-Rad 1610744). The proteins were then transferred onto 0.2-μm nitrocellulose membranes using a Bio-Rad Criterion blotter (Wet transfer) for 20 min at 100 V. The membrane was then incubated with 2% formaldehyde in PBS for 30 min to fix Aβ42 peptide to the membrane. Next, the membranes were blocked with 5% milk in 1xTBST for 1 hr at room temperature and incubated at 4 °C overnight with 6E10 antibody (BioLegend, Cat. No. 80300) and anti-apoE primary antibody (Abcam, Cat. No. ab183597) at 1:1000 and 1:2000 dilution, respectively. Secondary antibodies, Alexa Fluor® 680 Anti-Rabbit IgG (Jackson ImmunoResearch Laboratories, Cat. No. Inc 711-625-152), and Alexa Fluor® 790 Anti-Mouse IgG (Jackson ImmunoResearch Laboratories, Cat. No. 715-655-150) respectively were used at 1:50,000 dilution in 1xTBS-T (0.1%

tween-20 in tris-buffered saline) for 1 hr at room temperature to stain the membranes. The membranes were washed with TBS-T for 10 min three times after primary and secondary antibodies incubation. Finally, proteins were visualized with an Odyssey XF Imager (LI-COR).

## Co-immunoprecipitation and western-blot assay
We performed co-IP assays using the Dynabeads Kit for Immunoprecipitation (ThermoFisher, Cat. No. 80105 G). Briefly, 50 μL (1.5 mg) of protein-G-coated Dynabeads were incubated with 5 μg of 6E10 antibody for 1 h at room temperature. After incubation, the Dynabeads were washed once with 200 μL washing buffer (provided with the kit) and then mixed with 100 μg of the Aβ42 samples. The mixture was rotated overnight at 4 °C to immunoprecipitate Aβ42. Following this incubation, the mixture was washed three times with the 200 μL washing buffer (provided with the kit) to remove non-specifically bound proteins. Then the mixture was placed on the magnet and supernatant was removed. Then 20 μL elution buffer (provided with the kit) was added and gently mixed with pipette to resuspend the magnetic bead- immunoprecipitate complex. Then it was incubated for 2 min with rotation at room temperature to dissociate the complex from the magnetic beads. Then the mixture was again placed on the magnet and the supernatant containing immunoprecipitate transferred to a new low-binding tube. Next, the proteins in immunoprecipitate were separated by electrophoresis using 15% Tris-Glycine sodium dodecyl sulfate (SDS) polyacrylamide gels. Proteins were next transferred to nitrocellulose membranes using the Trans-Blot Turbo Transfer System (BioRad) and Trans-Blot Turbo Midi 0.2 μm Nitrocellulose Transfer Packs (BioRad) LOW MW programme, optimized for more efficient transfer of small proteins, for 5 min. The 0.2 μm nitrocellulose membrane was then blocked in 5% skimmed milk in 1xTBS-T for 1 hr at room temperature and incubated at 4 °C overnight with the 6E10 antibody (BioLegend, Cat. No. 80300) and anti-apoE primary antibody (AbCam, Cat. No. ab183597) at 1:4000 and 1:1000 dilution, respectively. Secondary antibodies, Alexa Fluor® 680 Anti-Rabbit IgG (Jackson ImmunoResearch Laboratories, Cat. No. Inc 711-625-152), and Alexa Fluor® 790 Anti-Mouse IgG (Jackson ImmunoResearch Laboratories, Cat. No. 715-655-150) respectively were used at 1:50,000 dilution in 1xTBS-T (0.1% Tween-20 in TBS) for 1 hr at room temperature to stain the membranes. The membranes were washed three times with TBS-T for 10 min after primary and secondary antibodies incubation. Finally, proteins were visualized with an Odyssey XF Imager (LI-COR).

## Human iPSC-derived microglia-like (iMGL) Culture and immunocytochemistry
The iMGL differentiation protocol assays were performed following previously described methodology[45,46]. Induced pluripotent stem cells (iPSCs) (female donor, 52 years old) were maintained using mTeSR+ medium (StemCell Technologies, Cat. No. 85850) on vitronectin-coated plates until they reached approximately 80% confluence. They were then passaged with Relesr (StemCell Technologies, Cat. No. 100-0484) and seeded onto Matrigel-coated plates (StemCell Technologies, Cat. No. 100-0484) at 140,000 cells per 10 cm². To induce differentiation, iPSCs were exposed to E8 media (StemCell Technologies, Cat. No. 05990), supplemented with 1% penicillin-streptomycin (ThermoFisher Cat, No. 15140122, 10 μM Rho kinase inhibitor (ROCKi) (StemCell Technologies, Cat, No. 72304), 5 ng/ml BMP4 (Peprotech, Cat. No. 120-05ET), 1 μM CHIR99021 (Axon, Cat. No. 1386), and 25 ng/ml activin A (Peprotech, Cat. No. 120-14 P). The cells were maintained under these conditions for 24 h at 37 °C in an atmosphere of 5% $O_2$ and 5% $CO_2$. After the initial 24 h, we replaced the media with a fresh batch of the same composition, but with reduced concentration (1 μM) of ROCKi, and continued incubation under the same environmental conditions. At the 44 h mark post-plating, the cells were

transitioned to FVI media. The FVI media comprised of DF3S media (consisting of DMEM/F12 (ThermoFisher, Cat. No. 11320033), 1X GlutaMAX (ThermoFisher, Cat. No. 35050038), 0.5% penicillin-streptomycin (Thermo Fisher, Cat. No. 15140122), 64 mg/L L-ascorbic acid (Sigma, Cat. No. A4403), 14 μg/L sodium selenite (Sigma, Cat. No. S5261), 543 mg/L sodium bicarbonate (Sigma, Cat. No. S6014)) supplemented with 200 ng/mL FGF2 (Peprotech, Cat. No. 100-18B), 10 μM SB431542 (StemCell technologies, Cat. No. 72232), 5 μg/mL insulin (Sigma, Cat. No I9278), and 50 ng/mL VEGF (Peprotech, Cat. No. 100-20). This media was replaced with fresh FI media again after 24 h. After another 24 h period, we shifted the cells to normoxic conditions. The cell media was replaced with HPC media, comprised of DF3S media (as described above), supplemented with 50 ng/mL FGF2, 5 μg/ml insulin, 50 ng/mL VEGF, 50 ng/mL TPO (StemCell Technologies, Cat. No. 78210), 10 ng/ml SCF (StemCell Technologies, Cat. No. 78155), 50 ng/mL IL-6 (StemCell Technologies, Cat. No. 78148) and 10 ng/mL IL3 (StemCell Technologies, Cat. No. 78146). This HPC media was replaced every 24 h for the next 4 days until distinct cobblestone patches of cells began to appear. As these cells started to bloom, we collected the media from these dishes, centrifuged it at 300 g for 5 min, and re-suspended the resultant pellet in fresh HPC media. This suspension was then reintroduced to the original wells. After 24 h, we collected the blooming progenitors, which could be identified by their round and bright appearance. This collection process involved gently dislodging lightly attached cells using a P1000 pipette, filtering them through a 70 μm cell strainer, and centrifuging them at 300 g for 5 min. The cells were then seeded at a density of 500,000 cells per 10 cm² in ultra-low attachment dishes (Corning Cat. No. 16855831) designed to discourage attachment, containing Proliferation media (PM). PM is comprised of IMDM media (Thermo-Fisher, Cat. No. 12440053) supplemented with 1% penicillin-streptomycin, 10% FBS (ThermoFisher, Cat. No. 16000044) and 5 μg/mL insulin, 5 ng/mL MCSF (StemCell Technologies, Cat. No. 78150) and 100 ng/mL IL-34 (StemCell Technologies, Cat. No. 100-0930). After 48 h in PM media, the cells were collected, centrifuged at 300 g for 5 min, and the supernatant was discarded. The remaining pellet was resuspended, and these cells were then reseeded at the same density as before in poly-D-lysine coated 96 well plates (Phenoplate Parkin Elmar Cat. No. 6055508). Following seeding, they were nourished with fresh media twice daily. By day 16, the cells were ready for further experiments.

For the uptake experiments involving iMGLs, 25 μL of media was removed from each well (out of a total volume of 100 μl each well), and 25 μL of 4 μM Aβ (co-)aggregates in PBS were added, resulting in a final concentration of 1 μM monomer equivalent. The cells were then incubated for 1 h. After this incubation, the medium was removed and snap-frozen in liquid nitrogen, and stored at -80 °C for further analysis. The cells were washed with 1x PBS and fixed using 4% formaldehyde for 10 min at room temperature. After fixation, the cells were washed three times with 1x PBS and permeabilized in 0.3% Triton-X 100 (Sigma, Cat. No. T8787) in PBS for 5 min at ambient temperature. The cells were then blocked for 1 h in 3% BSA-PBS, followed by three washes with PBS. Subsequently, the cells were incubated overnight with a mixture of 60 nM recombinant Anti-Iba1 antibody [EPR16588] (Abcam Cat. No. ab178846), and 25 nM Aβ specific 6E10 (Biolegend Cat. No. 803001) in PBS with 1% BSA and 0.1 mg/mL horse serum (Thermo-Fisher Cat. No. 16050122). Iba1 staining was used to define the boundary of cells and DAPI for nuclear staining. Next, secondary antibodies Alexa Fluor 488 labelled Goat anti-Mouse IgG (Thermo-Fisher Cat. No. A-11029) and Alexa Fluor 647 labelled Donkey anti-Rabbit IgG, (ThermoFisher Cat. No. A-31573) were introduced at a concentration of 1:1000 for 1 h. These iMGLs were then washed three times with PBS, Then 1 ug/mL of DAPI (ThermoFisher Cat. No. 62248) in PBS was used for nuclear staining and then cells washed three times

with PBS, Finally the iMGLs were imaged using the Opera Phenix High Content Imaging System (PerkinElmer®).

For iMGLs staining, microglia specific markers, Iba1 and CD45, were employed. The Anti-Iba1 antibody [EPR16588] (Abcam, Cat. No. ab178846) was applied at a concentration of 1:250, while the Recombinant Anti-CD45 antibody [EP322Y] (Abcam, Cat. No. ab40763) was used at a concentration of 1:100) in PBS with 1% BSA and 0.1 mg/mL horse serum. To facilitate the detection of these rabbit primary antibodies, Alexa Fluor 488-labelled Donkey anti-Rabbit IgG (Thermo-Fisher, Cat. No. A-21206) was used at a concentration of 1:1000 in PBS.

## Human iAstrocyte culture and immunocytochemistry

The iAstrocyte differentiation protocol assays were performed following previously described protocol[47,48]. To establish fibroblast cultures, we initially plated fibroblasts obtained from healthy controls (female, age 61) at a density of 250,000 cells per well. After a 24 h incubation, we introduced retroviral vectors (OCT3, Sox2, KLF4 and C-MYC for 12 h to transduce the cells. Following 48 h of transduction, we transitioned the cells to an iNPC (induced neuronal progenitor cell) medium, comprising DMEM/F12 (ThermoFisher Gibco Cat. No 11320033), 1% N-2 supplement (ThermoFisher Gibco Cat. No 17502048), 1% B27 (ThermoFisher, Cat. No. A3653401), EGF (40 ng/ml) (Peprotech, Cat. No. AF-100-15), and FGF (20 ng/ml) (Peprotech, Cat. No. 100-18B). To differentiate iNPCs into iAstrocytes, we seeded iNPCs onto a 10 cm dish coated with fibronectin (5 μg/ml, R&D system 1918-FN) in DMEM media (ThermoFisher, Cat. No. 11320033) containing 10% FBS (ThermoFisher, Cat. No. 16000044) and 0.3% N-2 supplement for 7 days. For subsequent experiments involving iAstrocytes, we plated them onto a black 96-well plate (Greiner bio-one) at a density of 2500 cells per well.

These iAstrocytes were then exposed to Aβ (co-)aggregates. Each well had 25 μL of its media removed from a total volume of 100 μL, and it was substituted with 25 μL of 4 μM Aβ (co-) aggregates in PBS, which resulted in a final concentration of 1 μM Aβ42 monomer equivalent. Subsequently, the cells underwent a 1 h incubation period. Following the incubation, the medium was removed, snap-frozen in liquid nitrogen, and then stored at −80 °C for subsequent analysis. Then, we fixed the iAstrocytes with 4% formaldehyde for 10 min immediately following the 1 h exposure to Aβ (co-)aggregates. Subsequently, we permeabilized the cells using a solution of 0.1% Triton-X100 (Sigma-Aldrich Cat. No. X100-100ML) in PBS-T (0.1% Tween-20 in PBS) for 30 min, followed by two washes with PBS-T. To minimize non-specific binding, we incubated the cells in PBS-T with 5% horse serum for 60 min. Primary antibodies against CD44 (Abcam, Cat. No. ab157107), and Aβ specific 6E10 (Biolegend Cat. No. 803001) in PBS with 1% BSA, were applied and allowed to incubate overnight. Subsequently, secondary antibodies Alexa Fluor 488 labelled Goat anti-Mouse IgG (ThermoFisher Cat. No. A-11029) and Alexa Fluor 647 labelled Donkey anti-Rabbit IgG, (ThermoFisher Cat. No. A-31573) were introduced at a concentration of 1:1000 for 1 h, followed by staining with Hoechst (Life Technologies, Cat. No. H3570) at a concentration of 2.5 μM for 10 min. The iAstrocytes were then rinsed three times with PBS, and the internalized Aβ42 was imaged using the Opera Phenix High Content Imaging System (PerkinElmer).

In our iAstrocyte staining protocol, we utilized established astrocytic markers including vimentin, CD44, and the glutamate transporter EAAT2. The antibodies applied were as follows: Anti-vimentin (Merck Millipore Cat. No. AB5733) at a 1:1000 dilution, Anti-CD44 (Abcam Cat. No. ab157107) at a 1:200 dilution, and Anti-EAAT2 (Abcam Cat No. ab41621) at a 1:250 dilution. These antibodies were diluted in PBS containing 1% BSA and 0.1 mg/mL horse serum. For the detection of Vimentin, we used Alexa Fluor 488-labelled Goat anti-Chicken IgY (H + L) Secondary Antibody (ThermoFisher Scientific, Catalog No. A-11039) at a 1:1000 concentration. For both CD44 and

EAAT2, Alexa Fluor 488-labelled Donkey anti-Rabbit IgG (Thermo-Fisher Scientific, Catalog No. A-21206) was employed, also at a 1:1000 concentration in PBS.

## Data analysis of cellular uptake assay

The data analysis was performed using Harmony High-Content Imaging and Analysis Software. In each experimental condition, images were taken from ten randomly selected fields of view, totaling approximately 60-100 cells per well across three wells. For image acquistion, three channels were used: 405 nm excitation for nuclear staining, 488 nm excitation for visualizing cell areas, with Iba1 staining for iMGLs and CD44 staining for iAstrocytes and 647 nm excitation for detecting Aβ42 antibody labeled with Alexa-Fluor-647. The analysis involved defining cell boundaries for iMGLs and iAstrocytes using Iba1 and CD44 staining, respectively. Masks were created to distinguish between cell and non-cell areas based on the 488 nm excitation images. These masks were then applied to the Aβ detection channels to isolate specific regions within the cells and measure fluorescence intensities, allowing for the quantification of Aβ uptake. Background signals were selected based on images of cells which were stained with secondary antibodies only and automatically subtracted from the original image.

## ELISA to measure cytokine and chemokine concentrations in cell media

To assay cytokine and chemokine secretion by iMGL cells and iAstrocytes, cell media were collected after incubation of cells with different aggregates for 1 h and stored at −80 °C until analyzed. MCP-1, IL-1β, IL-6 and TNF-α in the cell media was measured using the Duoset® enzyme-linked immunosorbent assay (ELISA Cat. No DY279, DY201, DY206, DY210) development system (R&D Systems, Abingdon, UK), according to the manufacturer's protocol.

## Membrane permeabilization assay

The membrane permeabilization assays were performed by following previously described methods[56]. Briefly, vesicles (mean diameter of 200 nm) were prepared by extrusion and five freeze-thaw cycles of a 100:1 mixture of phospholipids 16:0–18:1PC (Avanti Lipids, Cat. No. 850457 C) and biotinylated lipids 18:1-12:0 Biotin PC (Avanti Lipids, Cat. No. 860563 C). The lipid mixture was hydrated in 100 μM Cal-520 dye (Stratech, Cat. No. 21141) dissolved in HEPES buffer (50 mM, pH 6.5). Non-incorporated Cal-520 dye was separated from the dye-filled vesicles using size-exclusion chromatography, using a SuperdexTM 200 Increase 10/300 GL column attached to an AKTA pure system (GE Life Sciences) with a flow rate of 0.5 mL/min. Dye-filled vesicles were immobilized on coverslips (VWR International, 22 × 22 mm, 631-0122), which were first cleaned using an argon plasma cleaner (PDC-002, Harrick Plasma) for ~45 min to remove any organic and fluorescent impurities. Frame-seal incubation chambers (9 x 9mm, Biorad, SLF-0601) were then affixed to the coverslips, and 50 μL of a mixture of 100:1 PLL-g-PEG (20 kDa PLL grafted with 2 kDa PEG and 3.5 Lys units/PEG Chains, SuSoS AG) and PLL-g-PEG biotin (20 kDa PLL grafted with 2 kDa PEG and 50% 3.4 kDa PEG-Biotin, SuSoS AG) at ~1 mg/mL was added to the coverslips and incubated for ~30 min. The coverslips were then washed three times with HEPES, and 50 μL of 0.1 mg/mL NeutrAvidin (ThermoFisher, Cat. No. 31000) were added and incubated for 15 min. The coverslips were then rewashed three times with HEPES buffer, and 50 μL of the purified vesicles were added to the coverslips. Before imaging, 50 μL of Ca²⁺-containing buffer (phenol red-free Leibovitz's L-15 Medium, ThermoFisher Cat No. 21083027) was added, and blank images ($F_{blank}$) were recorded. The immobilized vesicles were then incubated with 50 μL of the aggregation mixture (100 nM in monomer equivalents) for 20 min and then reimaged ($F_{sample}$). Finally, 1 mg/mL of ionomycin (Cambridge bioscience, Cat. No. 1565-5) - which leads to Ca²⁺ saturation inside the vesicles - was added, and the same

areas were reimaged ($F_{ionomycin}$). The relative influx of $Ca^{2+}$ into an individual vesicle due to protein aggregates was then determined using the following equation: $Ca^{2+}$ influx = ($F_{sample}$ - $F_{blank}$)/($F_{ionomycin}$ - $F_{blank}$). The average degree of $Ca^{2+}$ influx was calculated by averaging the $Ca^{2+}$ influx in >200 individual vesicles. For the imaging, the Cal-520 dye was excited at 488 nm, and the emission from the dye was passed through a combination of a long-pass filter (BLP01-488R-25, Laser 2000) and a bandpass filter (FF01-520/44-25, Laser 2000) before being imaged using a Photometrics Evolve EMCCD camera. Images were acquired with a ~ 10 W/cm² power density with a scan speed of 20 Hz and bit depth of 16 bits.

### LDH cytotoxicity assay

The assay was performed using an LDH Cytotoxicity Assay Kit (Thermo Fisher, Cat. No. C20303) using human neuroblastoma SH-SY5Y (ATCC) cells cultured in DMEM media without Phenol Red (Gibco, Thermo Fisher, Cat. No. 21063029) and supplemented with 10% Fetal Bovine Serum (Gibco, Thermo Fisher, Cat. No. A4766801). Cells were then treated with aliquots of the aggregation reactions at $t_1$ and $t_3$ time points (final Aβ42 concentration: 2 μM in monomer equivalents) for 12 h, and the cell supernatant was assayed for LDH. The supernatant from cells treated with RIPA lysis buffer (Thermo Fisher, Cat. No. 89900) was used as a positive control, whereas media of the untreated SH-SY5Y cells were taken as a negative control. Next, the supplied reaction mixture (100 μL), which contains a substrate for the LDH activity detection reagent, was added to the medium according to the manufacturer's protocol. The reactions were quenched after 30 min with the Stop buffer (also supplied with the kit), and the absorbance at 480 nm was measured using a plate reader (Clariostar -BMG Biotech).

### Immunoprecipitation assay

Twenty-five microliters of Dynabeads Protein G (Invitrogen, Cat. No. 10007D) were mixed with 100 μL of a 100 nM solution of either HAE-4[16] or Mouse Isotype IgG2a kappa monoclonal [MG2a-53] (Biolegend) antibody in PBS and mixed in a tube rotator for 30 min at room temperature. The beads were then separated using a magnetic separation rack (DynaMag™-2 Magnet, Thermo Fisher, Cat. No. 12321D) and washed twice with PBS. Then, 300 μL of brain extract from *APOE4/4* homozygous AD patients or a mixture of in vitro-prepared lipidated and non-lipidated apoE-Aβ (150 μL each), from time-point $t_1$ were combined with the beads. This mixture was then rotated at 4 °C for 3 h. After that, the magnetic rack was used to separate the beads and collect the supernatant for imaging and cellular assays.

### Statistical tests

Data are displayed as the mean and error bars represent the standard deviation across replicates respectively. To assess the statistical significance of the differences between two populations we used unpaired two-sample t-tests. Multiple groups were compared using one-way ANOVAs with post hoc Tukey-tests. Values of $p < 0.05$ were considered statistically significant. The number of independent biological replicates and independent replicates for each experiment as well as details of statistical tests for each experiment can be found in figure legends. Statistical tests were calculated with Origin 9.0 (OriginLab).

### Reporting summary

Further information on research design is available in the Nature Portfolio Reporting Summary linked to this article.

### Data availability

All data are available from the corresponding authors upon request. Source data are provided with this paper.

### Code availability

Custom scripts used to run the cell uptake and colocalization analyses described in this study are available vis Github. https://github.com/zengjiexia/CellIntakeAnalysis.

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

## Acknowledgements

This study was supported by the EPSRC grant 2594676 (HEW, SD), Alzheimer's Society PhD studentship grant 396 (MM,SW), AMS grants SGL028\1097 (SMB), Parkinson's UK grants F-1301 (H.M.), NIH grants AG047644 (D.M.H.) and NS090934 (D.M.H.), European Research Council Grant Number 669237 (D.K.), the Royal Society (D.K.), Dementia Research UK Pilot Award (S.D.), the UK Dementia Research Institute (DRI) at Cambridge (D.K.), Academy of Medical sciene springboard award SBF006\1038 (S.D.) and a UKRI Future Leaders Fellowship (Grant number MR/V023861/1) (A.O., A.U., and S.D.). The tissue for this study was provided by the Newcastle Brain Tissue Resource, which is funded in part by a grant from the UK Medical Research Council and in part by Brains for Dementia Research, a joint venture between Alzheimer's Society and Alzheimer's Research UK. We thank Sneha Rajan and Guy Soane for helping with western blot study. We also thank the Professor Maria Spillantini, University of Cambridge laboratory for providing the facilities and expertise for processing of the brain tissue used in these experiments. The Sheffield NIHR Biomedical Research Centre provided support for this study.

## Author contributions

Z.X., E.E.P., A.O., R.T.R., and S.D. did the SiMPull experiments. R.T.R. and E.E.P. modified antibodies and coated coverslips for SiMPull. Data analysis and statistics for SiMPuLL were developed, carried out by Z.X., and interpreted by Z.X. and S.D., Z.X., E.E.P., H.E.W., and S.D. aggregated proteins. The cellular uptake assay was carried out by E.E.P., A.U., H.E.W., M.C.K., and E.E.P. and H.E.W. did the uptake data analysis and A.H. helped with the data analysis. E.E.P. and A.U. did the immunoprecipitation and western blot experiments and cytokine and chemokine measurement, H.D. and S.D. prepared soaked-brain extracts. Z.X., E.E.P., A.U., and S.D. did the membrane permeabilization and LDH assays. H.E.W., M.M. and S.M.B. carried out the astrocyte culture and M.C.K., S.O., M. G-B. carried out the microglia culture. T.L., K.A.B., A.O., E.Z., E.D., Y.L. J.L., Y.P.Z., and J.S.H.D. helped with antibody modifications, slide preparation, cell culture or microscopy. M.R.S. and H.J., and D. M. H. provided HAE-4 antibody and protocol of apoE lipidation. L.F., S.W., H.M., P.T., D.C.C., D.M.H., T.M., R.T.R., D.K., and S.D. supervised the project. Z.X., E.E.P., A.U., R.T.R., and S.D. were performed data analysis,interpretation of study data and make figures of the manuscript. The initial draft of the paper was written by E.E.P., A.U., R.T.R., D.K., and S.D.; all other authors critiqued the output, provided feedback and contributed in editing the manuscript into its final form. R.T.R., D.K., and S.D. designed and S.D. conceived the study. All authors read and approved the manuscript.

## Competing interests

D.M.H. is an inventor on a patent licensed by Washington University to NextCure on anti-apoE antibodies. D.M.H. co-founded and is on the scientific advisory board of C2N Diagnostics, DenaliGenentech, and Cajal Neuroscience. D.M.H. consults for Asteroid. The lab of D.M.H. receives research grants from the National Institutes of Health, Cure Alzheimer's Fund, Tau Consortium, the JPB Foundation, Good Ventures, the Rainwater Foundation, NextCure, Eli Lilly, and Ionis. D.C.C. and P.T. hold stock in AstraZeneca. All the other authors declare no competing interests.

## Additional information

[1]Yusuf Hamied Department of Chemistry, University of Cambridge, Cambridge, UK. [2]UK Dementia Research Institute at University of Cambridge, Cambridge, UK. [3]Sheffield Institute for Translational Neuroscience, Division of Neurosciences, University of Sheffield, Sheffield S10 2HQ, UK. [4]Clinical Neurosciences, University of Cambridge, Cambridge CB2 0QQ, UK. [5]SUPA School of Physics and Astronomy, University of St Andrews, North Haugh, St Andrews KY16 9SS, UK. [6]Department of Neurology, Hope Center for Neurological Disorders, Knight ADRC, Washington University School of Medicine, St. Louis, MO, USA. [7]Department of Pathology and Immunology, Washington University School of Medicine, St. Louis, MO, USA. [8]Neuroscience, Bio-Pharmaceuticals R&D, AstraZeneca, Cambridge, UK. [9]A.I. Virtanen Institute for Molecular Sciences, University of Eastern Finland, Kuopio, Finland. [10]Neuroscience Institute, University of Sheffield, Sheffield S10 2TN, UK. [11]Healthy Lifespan Institute (HELSI), University of Sheffield, Western Bank, Sheffield S10 2TN, UK. [12]These authors contributed equally: Zengjie Xia, Emily E. Prescott, Agnieszka Urbanek. ✉e-mail: rr360@cam.ac.uk; dk10012@cam.ac.uk; S.De@sheffield.ac.uk

