## [Peer Review File · Nature Communications]

Co-aggregation with Apolipoprotein E modulates the function of Amyloid- β in Alzheimer's diseaseEditorial Note: This manuscript has been previously reviewed at another journal that is not operating a transparent peer review scheme. This document only contains reviewer comments and rebuttal letters for versions considered at *Nature Communications*.

Reviewer #1 (Remarks to the Author):

The manuscript is greatly improved from previous iterations and the authors have patiently addressed in the rebuttal the points raised by the reviewers.

However, I think some of the remarks of Reviewer 2 should be explicitly addressed in the main text more obviously.

Reviewer 2 Remark 1. Difference in the response of A β uptake and inflammation for lipidated and non-lipidated apoE. The discussion is well argued and the main text would benefit of a larger mention of the discrepancy and contextualization of the results (some of the discussion can be inserted also as part of the supporting information)

Reviewer 2 Remark 2 and 3 (they can be combined). As above, I think the reader would benefit for a more thorough argumentation of the differences between isoforms and their critical interpretation. Reviewer 2 is correct in stating that trends are not obvious. On the other hand, the data presented by the authors are complex and very interesting, suggesting that different stage of oligomerization for A β and lipidation for apoE can lead to very different outcomes.

Corresponding tables of significance should be provided possibly in the Supporting Information or mentioned in Supporting Information and provided as Supporting data.

Reviewer #1 (Remarks to the Author):

The manuscript is greatly improved from previous iterations and the authors have patiently addressed in the rebuttal the points raised by the reviewers.

However, I think some of the remarks of Reviewer 2 should be explicitly addressed in the main text more obviously.

Comments: We greatly appreciate the reviewer's positive feedback on the improvements to our manuscript and their thorough review of our manuscript.

Reviewer 2 Remark 1. Difference in the response of A β uptake and inflammation for lipidated and non-lipidated apoE. The discussion is well argued and the main text would benefit of a larger mention of the discrepancy and contextualization of the results (some of the discussion can be inserted also as part of the supporting information)

Reviewer 2 Remark 2 and 3 (they can be combined). As above, I think the reader would benefit for a more thorough argumentation of the differences between isoforms and their critical interpretation. Reviewer 2 is correct in stating that trends are not obvious. On the other hand, the data presented by the authors are complex and very interesting, suggesting that different stage of oligomerization for A β and lipidation for apoE can lead to very different outcomes.

Comments: We appreciate the reviewer's careful review of our manuscript and are pleased that they found the data interesting. Based on remarks 1 and 2, we have added a paragraph to the discussion section of the main text to better explain these differences. The new paragraph can be found at lines 415-431.

Our work aims to address a significant gap in understanding by investigating how A β 42 interacts with various isoforms of apoE in both lipidated and non-lipidated states, and by assessing how these interactions affect disease-relevant functions of A β 42. We observed a stark contrast in the interactions between apoE and A β across the early (t_1) and later (t_3) stages of A β aggregation. Isoform-specific and lipidation-dependent differences in A β uptake and inflammation observed in early-stages (t_1) converged into a uniform response in later stages (t_3), indicating stage-specific interactions between A β and apoE. Furthermore, we observed functional differences when A β was associated with lipidated versus non-lipidated apoE. At the early stages of A β aggregation (t_1), glia-mediated clearance of A β was largely influenced by lipidated apoE, while the secretion of proinflammatory markers was predominantly induced by non-lipidated apoE. These findings highlight the critical influence of apoE lipidation at the early stages of A β aggregation. However, as the aggregation progresses to t_3 , the presence of apoE within the A β aggregates diminishes, along with its impact on A β function. Although there are slight variations in experimental trends, our study reflects the importance of considering the temporal aspects of apoE's influence on A β aggregation, clearance, and inflammation, which are crucial for understanding the mechanisms underlying AD pathogenesis.

Corresponding tables of significance should be provided possibly in the Supporting Information or mentioned in Supporting Information and provided as Supporting data.

Comments: We have added the corresponding tables of significance in Supporting Information